# Validation of the Cooling Model for TMCP Processing of Steel Sheets with Oxide Scale Using Industrial Experiment Data

Emmanuil Beygelzimer [1],* and Yan Beygelzimer [2]

1 OMD-Engineering LLC, 49000 Dnipro, Ukraine
2 Donetsk Institute for Physics and Engineering Named after A.A. Galkin, National Academy of Sciences of Ukraine, 03028 Kyiv, Ukraine; yanbeygel@gmail.com
* Correspondence: emmanuilomd@gmail.com; Tel.: +380-503686342

**Abstract:** To verify the mathematical model of the water-jet cooling of steel plates developed by the authors, previously performed experimental studies of the temperature of the test plates in a roller-quenching machine (RQM) were used. The calculated temperature change in the metal as it moved in the RQM was compared with the readings of thermocouples installed at the center of the test plate and near its surface. The basis of the model is the dependence of the temperatures of the film, transition and nucleate boiling regimes on the thickness of the oxide scale layer on the cooled surface. It was found that the model correctly accounts for the oxide scale on the sheet surface, the flow rates and combinations of the RQM banks used, the water temperature, and other factors. For all tests, the calculated metal temperature corresponded well with the measured one. In the experiments with interrupted cooling, the calculated temperature plots repeated the characteristic changes in the experimental curves. The main uncertainty in the modeling of cooling over a wide temperature range can be attributed to the random nature of changes in the oxide scale thickness during water cooling. In this regard, the estimated thickness of the oxide scale layer should be considered the main parameter for adapting the sheet temperature-control process. The data obtained confirm the possibility of effective application of the model in the ACS of industrial TMCP (Thermo-Mechanical Controlled Process) systems.

**Keywords:** rolled flat products; accelerated cooling; temperature; mathematical model; oxide scale; Thermo-Mechanical Controlled Process

## 1. Introduction

The current development of the global rolled steel market is characterized by a constant increase in the share of products produced by the Thermo-Mechanical Controlled Process (TMCP) [1–3]. The essence of this process lies in the combination of controlled hot rolling with controlled cooling in the temperature range of microstructural transformations in steel [4]. TMCP has the potential to impart such a combination of properties to rolled products (e.g., strength, ductility, toughness, cold resistance, weldability, etc.), which is not the case for other methods [5–9]. This feature is due to a special mechanism for the formation of fine-grained steel microstructures during phase transformations under rapid cooling conditions in combination with the deformed structure of the initial phase [4,10]. It is extremely difficult to achieve such a microstructure using other technological solutions (for instance, by microalloying) without controlled cooling. Moreover, in-line equipment for controlled cooling makes it possible to increase the efficiency of other technical and technological tools for imparting the required set of properties to the finished rolled products, including microalloying [11,12].

In practice, however, to benefit from TMCP, one must face the problems of ensuring precise control and uniformity of temperature and cooling rate, as well as flatness of the TMCP products [13]. When steel products are cooled with water, the biggest problems occur

when accelerated cooling to temperatures below about 550 °C occurs. This is due to the fact that during such cooling to these average mass temperatures, there is a sharp increase in cooling intensity as a result of the transition from the film to the nucleate regime of water boiling on the steel surface [14] (p. 424). The surface temperature at which this transition begins (often referred to as the Leidenfrost temperature) lies in a wide range—as a rule, from 300 to 800 °C—depending on a number of non-deterministic factors [15]; therefore, it can only be predicted with a fairly large error. In industrial rolling mills, the problem is exacerbated by the presence of oxide scale on the steel surface, since the thickness and properties of this scale have a very strong influence on the Leidenfrost temperature and heat transfer in general [16–22]. Therefore, the adequacy of the mathematical model of accelerated cooling in the presence of oxide scale on the cooled surface is critical for the efficient design and control of the TMCP process.

Several mathematical models used in various installations for the accelerated cooling of rolled steel are known from open sources [23–29]. However, these models do not take into account the thickness and properties of oxide scale on the surface of the cooled metal. Given these circumstances, the authors of this article have been developing their own mathematical model of the temperature evolution of steel sheets in a water-cooling unit of arbitrary configuration since the early 2000s. To date, the authors' model has been used in automated control systems on five industrial water-cooling units, including the 1700 hot strip mill, 2800 and 3000 plate mills, and roller-quenching machines [30–32].

The principal novelty of the latest version of the authors' model is that it explicitly takes into account the thickness and thermophysical properties of the oxide scale on the cooled surface. Verification and adaptation of individual structural components of the model were performed earlier based on the results of laboratory experiments, including those by other authors. However, a full assessment of the adequacy of the entire model can only be achieved via an active industrial experiment that allows one to naturally realize: the simultaneous cooling of the top and bottom surfaces of the sheet, changing of the heat-transfer mechanisms during cooling, and other features of real production conditions. Additionally, the most reliable data on the change in metal temperature can be obtained by measuring it directly inside the sheet, since the presence of oxide scale distorts the estimates of surface temperature.

On this basis, the goal of this work is to compare calculations according to the author's model with reliable measurements of the temperature inside the steel sheets with oxide scale during water cooling in industrial conditions, over a wide range of changes in the process parameters, including different regimes of water boiling.

## 2. Materials and Methods

### 2.1. Cooling Model

The authors' model implements numerical calculation of the temperature in the sheet as it moves along the cooling unit. Six different types of cooling zones are distinguished (Figure 1): (1) jet impingement, (2) supercritical parallel water flow on the top surface, (3) subcritical parallel water flow on the top surface, (4) parallel water flow on the bottom surface, (5) contact with a roll and (6) air cooling. Calculation of the heat-transfer coefficient in each zone is performed using physically meaningful (not statistical) equations. This makes it possible to take into account all the main physical factors determining the cooling process: the types of jets (circular or flat, compact or spray), the speed of water upon impingement and spreading on the sheet, the regimes of water boiling on the surface, the thickness of the oxide scale, the dependence of the thermophysical properties of steel, scale and water on temperature, etc.

Since the model is intended mainly for use in automated real-time TMCP control systems, the authors preferred simple engineering methods that eliminated repetitive lengthy calculations. That is why, in many cases, the authors were forced to find their own solutions, which, under specific conditions, produce results close to those of the rigorous but more time-consuming procedures. In other words, the authors followed an approach

to mathematical modeling, which does not involve developing as detailed a model as possible with a subsequent numerical solution, but selecting the main physical parameters that affect the process, and building a model that allows a simple analytical solution (the application of such an approach is described, for example, in [33,34]).

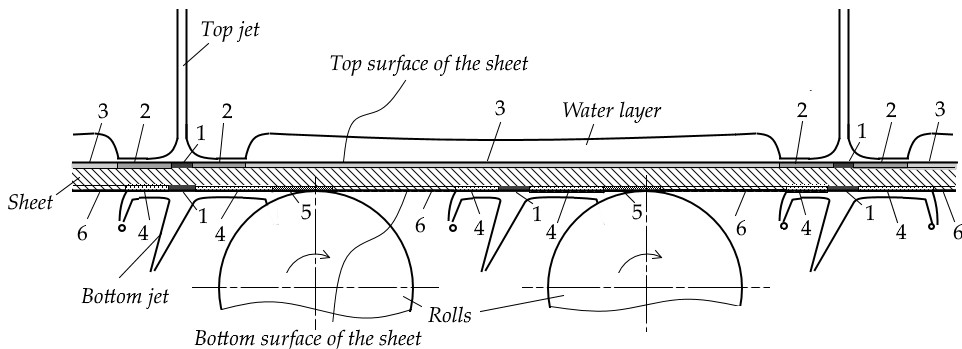

**Figure 1.** Schematic explaining the types of cooling zones considered in the model (indicated by numbers): 1—jet impingement; 2—supercritical flow on the top surface; 3—subcritical flow on the top surface; 4—parallel flow on the bottom surface; 5—contact with a roll; 6—air cooling.

The block diagram of the model is shown in Figure 2. Below is a brief description of the original authors' methods for calculating the key parameters. Known methods of other authors, used in the model under consideration, are given without a detailed description, only with appropriate references.

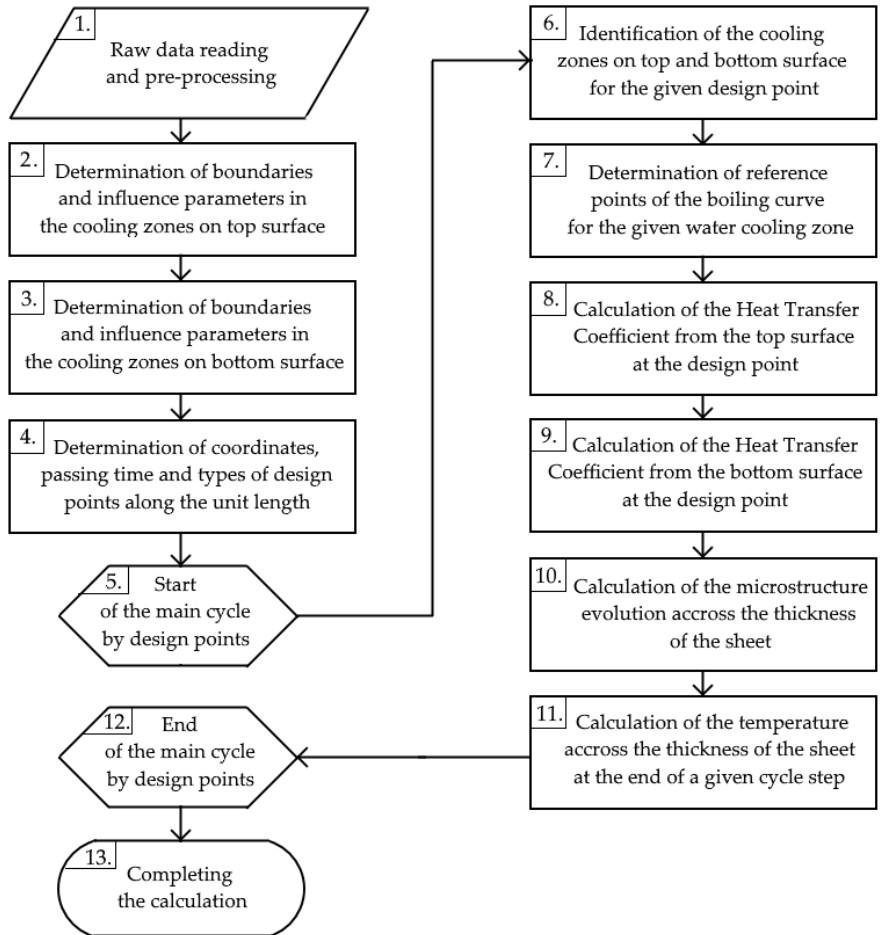

**Figure 2.** Block diagram of the cooling model.

2.1.1. Boundaries of Water-Cooling Zones and Parameters of Water Flow

Boundaries of water-cooling zones and parameters of water flow are used in blocks 2 and 3 of the diagram in Figure 2.

To determine the boundaries of an impingement zone produced by a spray jet (Figure 3), the authors, using analytical geometry methods, obtained the following formula for the *arbitrary radius of the impact spot* [35]:

$$R_\theta = \left( r_\theta + \frac{H}{\cos\gamma} \text{tg}\, \varphi \right) \frac{\sqrt{1 + (\sin\theta\, \text{tg}\gamma)^2}}{1 - \sin\theta \text{tg}\gamma\, \text{tg}\varphi} \tag{1}$$

where $\theta$ is a polar angle of the given radius such as the angle between this radius and the polar axis of the nozzle (the polar axis is parallel to the plane of the sheet and, in most cases, is aligned with the longitudinal axis of the collector with nozzles); $r_\theta$ is the nozzle outlet radius [m] with polar angle $\theta$; $R_\theta$ is the impact spot radius such as the length of the central projection of the given radius of the nozzle outlet on the sheet plane [m]; $H$ is the distance from the center of the nozzle outlet to the sheet plane [m]; $\gamma$ is an inclination angle such as the angle between the longitudinal axis of the jet and the perpendicular to the sheet plane; and $\varphi$ is an open angle such as the angle between the given generatrix of the spray cone and its longitudinal axis.

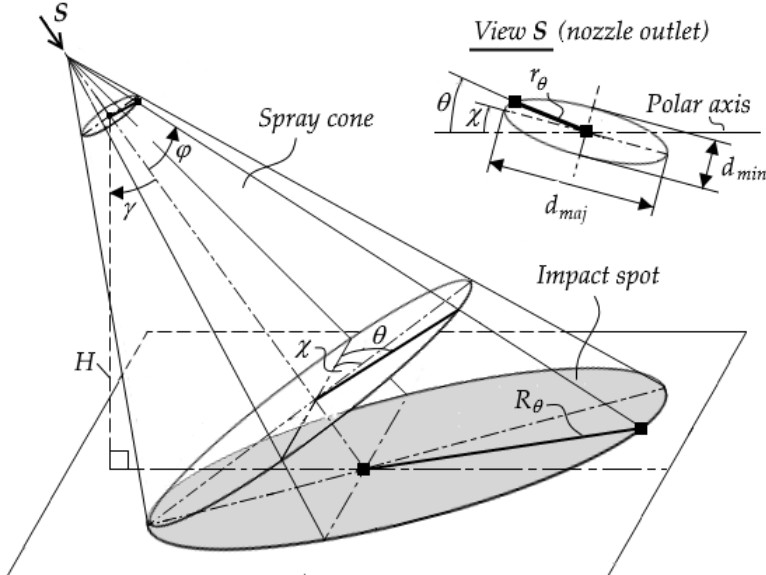

**Figure 3.** Schematic explaining the designations used in calculating the dimensions of the impact spot of the spray jet.

Based on Formula (1), the *area of the impact spot A* [m$^2$] is calculated as follows:

$$A = \frac{\pi}{4} \left( d_{\text{maj}} + \frac{2H}{\cos\gamma} \text{tg}\varphi_{\text{maj}} \right) \left( d_{\text{min}} + \frac{2H}{\cos\gamma} \text{tg}\varphi_{\text{min}} \right) k_{\text{maj}} k_{\text{min}} \tag{2}$$

where

$$k_{\text{maj}} = \frac{\sqrt{1 + (\sin\chi \text{tg}\gamma)^2}}{1 - \left( \sin\chi \text{tg}\gamma\, \text{tg}\varphi_{\text{maj}} \right)^2} \tag{3}$$

$$k_{\text{min}} = \frac{\sqrt{1 + (\cos\chi\, \text{tg}\gamma)^2}}{1 - \left( \cos\chi\, \text{tg}\gamma\, \text{tg}\varphi_{\text{min}} \right)^2} \tag{4}$$



$d_{\mathrm{maj}}$ and $d_{\mathrm{min}}$ are the major and the minor diameters of the nozzle outlet, respectively; $\varphi_{\mathrm{maj}}$ and $\varphi_{\mathrm{min}}$ are the open angles at the major and minor diameters of the nozzle outlet, respectively; $\chi$ is the nozzle rotation angle such as the angle between the polar axis and the major diameter of the nozzle outlet.

Compared to the known solution [36], Formula (2) takes into account the rotation of the nozzle around its longitudinal axis and the dimensions of the nozzle outlet orifice. Under the actual conditions of accelerated cooling systems, these factors can increase the impact area of the spray jet by up to 30%.

To calculate the *speed of the spray drops in the impingement zone*, the authors derived the approximate analytical solution of the differential equation of drop motion in the field of gravity, taking into account the resistance of the vapor–gas medium [37]. The key assumptions are the constancy of the inclination angle, the drag coefficient and the radius of the drop along its trajectory. These assumptions are valid for the operating conditions of spray nozzles for the accelerated cooling and quenching of metal sheets, namely: drop diameter $d = 0.5 \times 10^{-3} \ldots 3.0 \times 10^{-3}$ m; distance from the nozzle outlet to the cooled surface $H = 0.1 \ldots 1.0$ m; angle of the drop's departure from the nozzle relative to the horizontal $\beta_0 = 30 \ldots 90°$; and speed of the drop when departing from the nozzle $u_0 = 10 \ldots 40$ m·s$^{-1}$, Reynolds number Re $= u_0 d / v_a \approx 10^3 \ldots 10^4$ ($v_a$ is the kinematic viscosity of the vapor–gas medium [m$^2$·s$^{-1}$]). The final formula for the drop speed is as follows:

$$u = u_0 \exp(-k_u H S) \sqrt{1 \pm g \frac{\exp(2k_u H S) - 1}{u_0^2 k_u S}} \qquad (5)$$

where the "+" sign under the square root refers to the drop going downward (i.e., for the top jet), and the "−" sign refers to the drop going upward (i.e., for the bottom jet); $u$ is the speed of the water drop when impinging on the sheet surface [m·s$^{-1}$]; $u_0$ is the speed of the water drop as it leaves the nozzle [m·s$^{-1}$]; $k_u$ is the dimensionless speed parameter of the model (see below); $H$ is the distance from the nozzle outlet to the sheet plane [m]; $g$ is gravitational acceleration [m·s$^{-2}$]; $S$ is the complex parameter with the unit of [m$^{-1}$]:

$$S = \frac{3c\rho_a}{4d\rho} \qquad (6)$$

$c$ is the drag coefficient of the drop moving in a vapor–air medium (according to the data from [38] (p. 694), for a sphere, when $Re = 10^3 \ldots 10^4$, $c = 0.45$); $d$ is the diameter of the drop [m]; $\rho$ and $\rho_a$ are the density of the drop and the medium, respectively [kg·m$^{-3}$].

The analytical expressions for the speed parameter of the model $k_u$ are obtained via adaptation to the results of numerical integration of the differential equation of the motion for the top and the bottom drops separately:

— for the drops going downward:

$$\left.\begin{array}{ll} k_u = \frac{1}{\sin \beta_0}, & when \ \frac{H\sqrt{gS}}{u_0 \sin \beta_0} \leq 0.17 \\[2mm] k_u = \frac{1}{6}\left(1 + \frac{5}{\sin \beta_0}\right), & when \ \frac{H\sqrt{gS}}{u_0 \sin \beta_0} > 0.17 \end{array}\right\} \qquad (7)$$

— for the drops going upward:

$$k_u = \frac{1}{\sin \beta_0} \qquad (8)$$

where $\beta_0$ is the angle of the drop departure from the nozzle relative to the horizontal plane; for the axis of the spray jet, this angle is related to the jet inclination angle (see Figure 3) as $\beta_0 = \pi/2 - \gamma$.

The model allows us to calculate the *dimensions of the supercritical water flow zone* on the top surface (i.e., before the start of the hydraulic jump) for both circular and flat jets.

The supercritical flow zone from the *circular jet* is simplistically divided into two regions—inviscid and viscous flow (Figure 4)—with the assumption of constant water height in the viscous flow region. Based on these simplifications, solving the differential equation of gradually varied flow in an open segmental channel with no slope [39] (p. 119), together with the ordinary hydraulic jump equation [40] (p. 653), leads to the following results [41].

— The boundary post-jump height, above which the jump becomes submerged [m]:

$$h_{2\text{sub}} = \sqrt{\frac{a^2}{64} + \frac{V^2}{2g\pi^2 a^3}} - \frac{a}{8} \tag{9}$$

— The boundary post-jump height, above which the jump occurs inside the inviscid flow region [m]:

$$h_{2\text{vis}} = \sqrt{\frac{h_1^2}{4} + \frac{2V^2 h_1}{g\pi^2 a^4}} - \frac{h_1}{2} \tag{10}$$

where $V$ is the jet flow [m$^3$·s$^{-1}$]; $a$ is the radius of the undisturbed jet before impinging on the surface [m]; $g$ is gravitational acceleration [m·s$^{-2}$]; $h_2$ is the height of the water layer after the jump (post-jump height) [m]; $h_1$ is the height of the water layer before the jump (pre-jump height) in case the jump occurs inside the viscous flow region [m]:

$$h_1 = 200n^{1.69} a \tag{11}$$

$n$ is the roughness coefficient between water flow and sheet surface (for the considered conditions, $n$ values are between about 0.007 and 0.020, depending on the laminar or turbulent flow mode, the roughness of the sheet surface and the phase state of the water. Constants in Formula (11) are obtained via approximation of the numerical solution of the differential equation for the flow height).

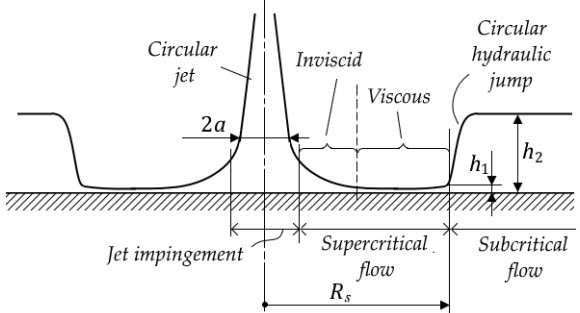

**Figure 4.** Schematic of the flow of a circular liquid jet on a horizontal plate with the formation of a circular hydraulic jump. The zones of jet impingement, and supercritical and subcritical flow are shown. Supercritical flow zone is conventionally divided into two regions: inviscid and viscous.

— *Radius $R_s$ of the circular hydraulic jump* [m] (subscript "s" denotes "spot"):
   (a)    if $h_2 \leq h_{2\text{vis}}$ (jump occurs inside the viscous region), then

$$R_s = \frac{V\sqrt{2}}{\pi} \left[ gh_2^3 \left( \left( \frac{2h_1}{h_2} + 1 \right)^2 - 1 \right) \right]^{-\frac{1}{2}} \tag{12}$$

   (b)    if $h_{2\text{vis}} < h_2 \leq h_{2\text{sub}}$ (jump occurs inside the inviscid region), then (expression (13) is known as the Rayleigh formula for an ideal fluid [42]).

$$R_s = \frac{V^2}{g\pi^2 h_2^2 a^2} - \frac{a^2}{2h_2} \tag{13}$$

In contrast to the known formulas [43], the above model allows us to calculate the circular jump radius without iterative procedures.

In order to obtain a simple engineering estimate of the location of a hydraulic jump in a *plane-parallel* flow (Figure 5), the authors of this article proposed to approximate the exponent $y$ in the following well-known Pavlovsky formula for the Chezy coefficient [44] (p. 91):

$$C = \frac{1}{n} h^y \tag{14}$$

where $C$ is the Chezy coefficient [$m^{1/2}s^{-1}$]; $n$ is the roughness coefficient (the physical meaning is the same as in Formula (11)); $h$ is the liquid-flow height [m]; $y$ is the exponent that, in its original form, depends on roughness coefficient $n$ and flow height $h$. Under conditions typical for industrial accelerated cooling systems ($h = 10^{-3} \dots 10^{-1}$ m; $n = 0.007 \dots 0.02$) the function $y = y(n, h)$ with an error of no more than 6% can be approximated by the function of only one variable—the roughness coefficient [45]:

$$y = 4.5n^{0.78} \tag{15}$$

According to (15), the exponent $y$, can be considered independent of the flow height $h$. With this assumption, the differential equation of gradually varied flow in an open wide rectangular channel with no slope can be solved in its explicit form [45]:

$$L_s = \frac{1}{n^2} \left[ \frac{\alpha}{g(1+2y)} \left( h_1^{1+2y} - h_0^{1+2y} \right) - \frac{1}{Q_f^2(4+2y)} \left( h_1^{4+2y} - h_0^{4+2y} \right) \right] \tag{16}$$

where $L_s$ is *the length of the supercritical zone* such as the distance from the border of the jet impact zone to the hydraulic jump toe [m]; $\alpha \approx 1{,}05$ is the kinetic energy correction factor; $Q_f$ is the specific flow rate per unit width [$m^2 \cdot s^{-1}$]; $h_0$ is the entry flow height such as the height at the border of the jet impingement zone [m]; and $h_1$ is the pre-jump height [m], which is related to the post-jump height by the known ratio [44] (p. 251):

$$h_1 = \frac{h_2}{2} \left( \sqrt{1 + \frac{8Q_f^2}{gh_2^3}} - 1 \right) \tag{17}$$

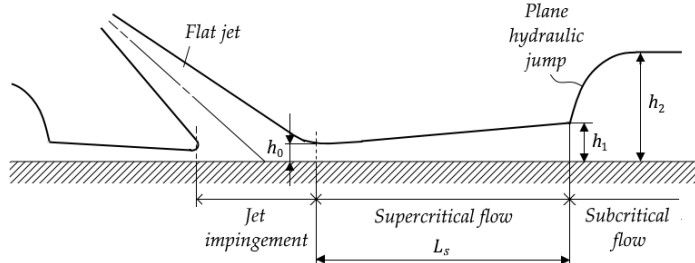

**Figure 5.** Schematic of the flow of a flat liquid jet on a horizontal plate with the formation of a plane hydraulic jump. The zones of jet impingement, supercritical and subcritical flow are shown.

The model considers four types of *subcritical flow zone*, i.e., the zone that lies beyond the hydraulic jump (Figure 6):

(a)  "clamped layer"—the water layer between closely spaced jets on a limited area of the surface. The height of such a layer may be so great that the jets are no longer able to overcome it (the hydraulic jump becomes submerged);

(b)　"bounded layer"—the water layer between adjacent rows of jets or between a row of jets and a pinch roll separated by a relatively large distance;

(c)　"open layer"—the water layer that spreads over the surface of the sheet without any obstruction; such a layer is formed, as a rule, before the first or after the last cooling bank in the absence of special devices for removing water from the surface;

(d)　"shifted layer"—the water layer which is removed from the sheet by special devices (hydro or pneumatic separators).

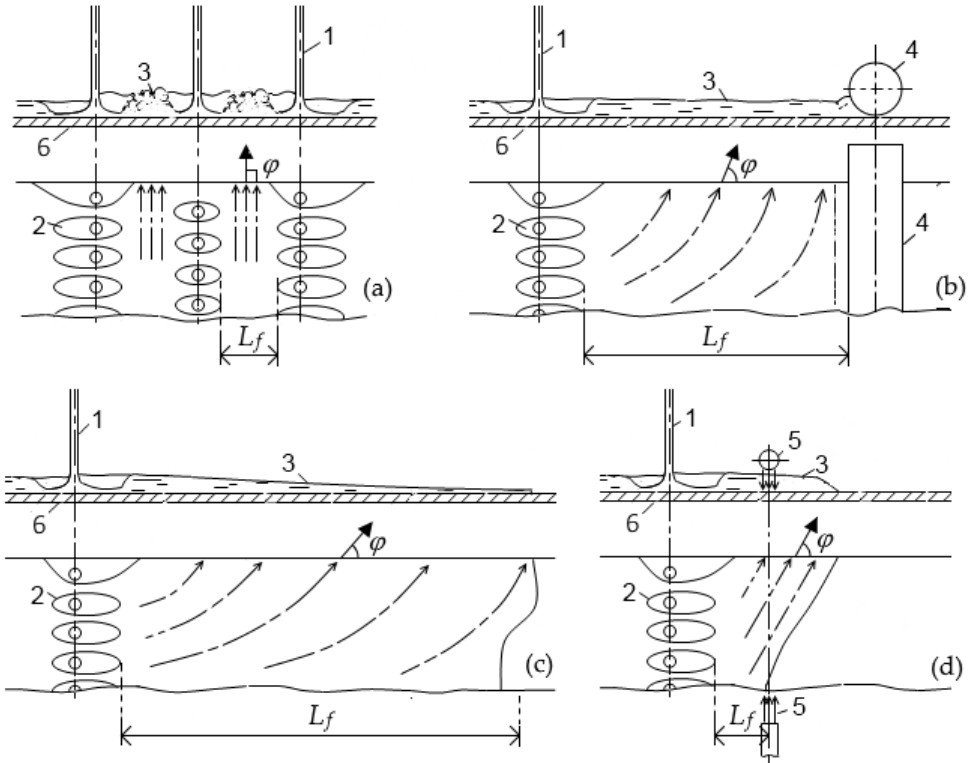

**Figure 6.** Different types of subcritical flow zone ("water layer"): (**a**) clamped, (**b**) bounded, (**c**) open and (**d**) shifted. Designations: 1—jet, 2—supercritical flow area, 3—water layer, 4—roll, 5—separator. $L_f$ is the length of the water layer; $\varphi$ is the runoff angle.

For each of the above types of water layer, the model calculates the height and the velocity of the flow; in addition, for the open layer type, it calculates the length of the subcritical zone $L_f$ [46,47]. The method developed is based on dividing the water layer into three regions (Figure 7): direct flow, oblique flow and stagnation. Assuming that the water flow varies gradually, the equations of motion are solved for the median streamline in the direct and oblique flow regions. As a boundary condition in the oblique flow region, the equality-to-unity of the Froude number on the lateral edge of the sheet is assumed (as on the weir threshold [48] (p. 391)).

In the direct-flow region, continuity of the layer height at the border with the oblique flow region is assumed as a boundary condition. At the same time, the discontinuity of velocity at this border is allowed.

As a result, the authors have proposed a procedure that allows for the calculation of the height and speed of the water layer at all reference points of the subcritical flow zone. The key parameter of this procedure is the "angular runoff coefficient" $k_\varphi$, which means the sinus of the runoff angle, i.e., $k_\varphi \equiv \sin \varphi$.

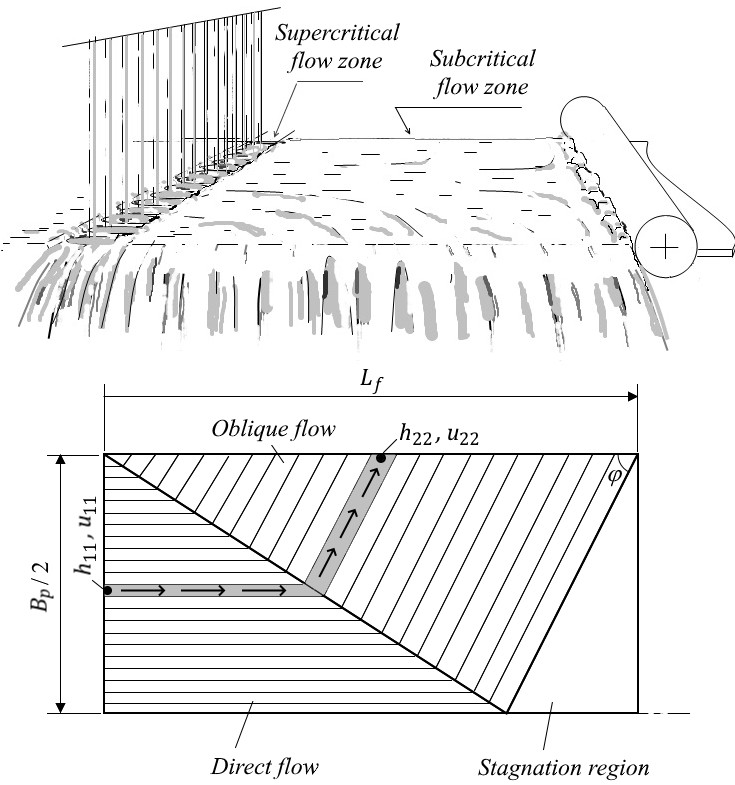

**Figure 7.** Schematic explaining the basic idea of calculating the parameters of the subcritical water flow zone by conventionally dividing this zone into three regions: direct flow, oblique flow and stagnation region. The arrows show the direction of water flow in the median streamline. The designations are in the text.

For a bounded layer, the calculation of $k_\varphi$ is performed using a chain of formulas:

— auxiliary parameter $\Phi$:

$$\Phi = n^{4/3} \left[ g^{2(4+2y)} \left( \frac{2L_f}{Q_f} \right)^{4(1+2y)} \right]^{1/9} \qquad (18)$$

where $n$ is the roughness coefficient (the physical meaning is the same as in Formula (11)); $g$ is gravitational acceleration [m·s$^{-2}$]; $Q_f$ is the specific flow rate per unit width [m$^2$·s$^{-1}$]; $L_f$ is the length of the subcritical zone [m]; and $y$ is the exponent (15);

— Friction parameter $\Omega_\Phi$:

$$\Omega_\Phi = (1.05 + 0.43\Phi)^{3/2} \qquad (19)$$

— Shape parameter $\delta$:

$$\delta = \frac{2L_f}{B_p} \qquad (20)$$

— Angular runoff coefficient

$$k_\varphi = \left( \frac{\Omega_\Phi}{\delta} \right)^x \qquad (21)$$

where $B_p$ is the width of the sheet [m]; $x = 0.5$ when $\Omega_\Phi/\delta < 1$; and $x = 0$ when $\Omega_\Phi/\delta \geq 1$.

With a known value of $k_\varphi$, the height and speed of the flow at all key points of the water layer can be calculated—for example,

$$h_{22} = \left( \frac{Q_f B_p}{2 k_\varphi L_f \sqrt{g}} \right)^{\frac{2}{3}} \tag{22}$$

$$u_{22} = \sqrt{g h_{22}} \tag{23}$$

where $u_{22}$ and $h_{22}$ are the speed [m·s$^{-1}$] and the height [m] of the water at the lateral edge, respectively (expression (23) follows from the equality of the Froude number-to-unity);

To calculate *the length of the subcritical flow zone $L_f$* in the case of the open water layer, the iteration procedure is realized. The essence of this procedure consists of solving, with respect to the desired length $L_f$, the system of algebraic equations obtained for the bounded layer (with the angular flow coefficient according to (21)), together with the following equation obtained for the open layer:

$$V_{fs} = k_m m'_s L_f \sqrt{2g} \left( h_{11} + \frac{u_{11}}{2g} \right)^{3/2} \tag{24}$$

where $V_{fs}$ is the volume of water that runs off from one lateral edge of the sheet per unit of time [m$^3$·s$^{-1}$]; $k_m = 0.15 \ldots 0.35$ is the model parameter; $h_{11}$ and $u_{11}$ are the height [m] and the speed [m·s$^{-1}$] of the water at the beginning of the subcritical zone, respectively; and $m'_s$ is the lateral spillway discharge coefficient [49]:

$$m'_s = 0.45 - 0.22 \frac{u_{11}^2}{g h_{11}} \tag{25}$$

### 2.1.2. Heat Flux in the Water-Cooling Zones

Heat flux in the water-cooling zones is used in blocks 7–9 of the diagram in Figure 2.

The calculation of heat flux in the water-cooling zones at any surface temperature is performed using the concept of reference points of the boiling curve [50], which allows us to avoid the breaking of solutions at critical points in the boiling regime change. According to this concept, on a boiling curve [51,52]—i.e., on the graph of the dependence of the heat flux on the surface temperature—in general, six reference points of different boiling regimes are distinguished (Figure 8):

DFB—Departure of Film Boiling;

EFB—End of Film Boiling (start of transient boiling);

DTB—Departure of Transient Boiling;

ETB—End of Transient Boiling (start of nucleate boiling);

DNB—Departure of Nucleate Boiling;

ENB—End of Nucleate Boiling (start of single-phase convection).

The heat flux at any surface temperature is calculated via linear interpolation between the reference points of the boiling curve. For example, if the current surface temperature refers to the transient boiling section, the heat flux is calculated as follows (the interpolation straight line for this section is shown in Figure 8):

$$q(t_w) = q_{\text{DTB}} + (t_w - t_{\text{DTB}}) \frac{q_{\text{EFB}} - q_{\text{DTB}}}{t_{\text{EFB}} - t_{\text{DTB}}} \tag{26}$$

where $q(t_w)$ is the heat flux as a function of the surface temperature [W·m$^{-2}$]; $t_w$ is the sheet surface temperature (subscript $w$ from "wall") [°C]; $t_{\text{DTB}}$ and $t_{\text{EFB}}$ are the values of the surface temperature at the reference points DTB (Departure of Transient Boiling) and EFB (End of Film Boiling), respectively [°C]; $q_{\text{DTB}}$ and $q_{\text{EFB}}$ are the values of the heat flux at the DTB and EFB reference points [W·m$^{-2}$].

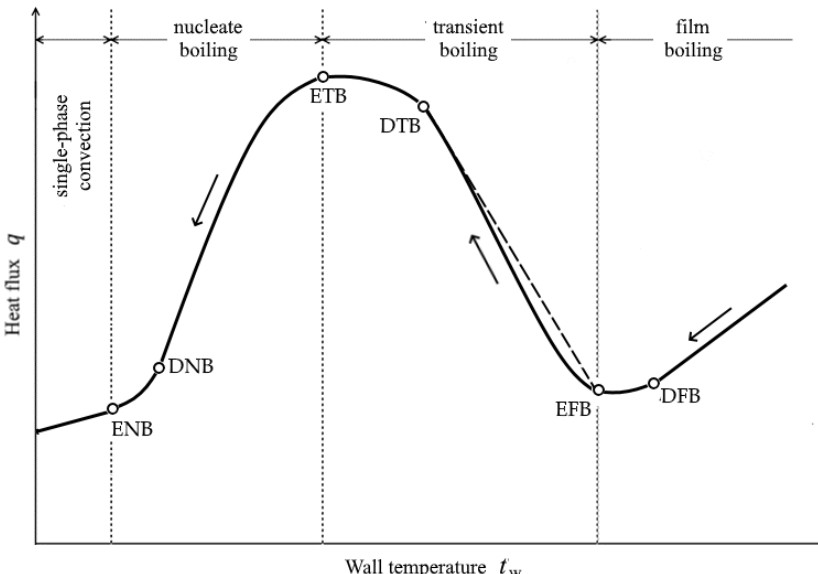

**Figure 8.** General view of the boiling curve. The deciphering of the reference points is given in the text. Arrows show the direction of parameter changes during cooling.

To implement the described approach, the coordinates (i.e., temperature and heat flux density) of all the reference points of the boiling curves for each water-cooling zone are calculated in the model.

The *End of Film Boiling temperature* $t_{\text{EFB}}$ [°C] is calculated by the formula proposed by the authors:

$$t_{\text{EFB}} = t_{int} + \left( t_{int} - t_f \right) \left( \frac{\rho_f c_f}{\lambda_w \rho_w c_w} \right)^{1/2} k_\tau^{1/2} \text{Re}^{1/3} \tag{27}$$

where $t_{int}$ is the temperature at the liquid–solid interface at the EFB point [°C]:

$$t_{int} = t_s + \xi \left( t_{f,lim} - t_s \right) \tag{28}$$

$t_s$ is the liquid saturation temperature [°C]; $t_f$ is the current liquid temperature away from the cooled surface [°C]; $t_{f,lim}$ is the practical limit of the maximum achievable liquid-phase temperature when impulse heating [°C] (for water at atmospheric pressure, we accepted $t_{f,lim}$ = 300 °C according to [53]; and $\xi$ is the subcooling effect coefficient. For water at atmospheric pressure:

$$\xi = 0.4 + 0.004 \cdot \Delta t_{sub} \tag{29}$$

$\Delta t_{sub}$ is the value of liquid subcooling [°C]:

$$\Delta t_{sub} = t_s - t_f \tag{30}$$

$\rho_f$ and $c_f$ are the density [kg·m$^{-3}$] and true isobaric specific-mass-heat capacity [J·kg$^{-1}$K$^{-1}$] of liquid at the temperature of $t_f$, respectively; $\lambda_w$, $\rho_w$ and $c_w$ are the thermal conductivity [W·m$^{-1}$·K$^{-1}$], density [kg·m$^{-3}$] and true isobaric specific-mass-heat capacity [J·kg$^{-1}$·K$^{-1}$] of the subsurface layer of the cooled sheet at the temperature of $t_w$, respectively (see below); Re is the Reynolds number in the form:

$$\text{Re} = \frac{2u_f}{v_f} \sqrt{\frac{\sigma_{fs}}{g\left(\rho_{fs} - \rho_{vs}\right)}} \tag{31}$$

$u_f$ is the liquid speed [m·s$^{-1}$]; $v_f$ is the kinematic viscosity of the liquid [m$^2$·s$^{-1}$]; $\sigma_{fs}$ is the surface-tension coefficient of liquid at the saturation temperature [N·m$^{-1}$]; $g$ is the

gravitational acceleration [m·s$^{-2}$]; $\rho_{fs}$ and $\rho_{vs}$ are the density [kg·m$^{-3}$] of liquid and vapor at the saturation temperature, respectively; and $k_\tau$ is the model parameter [W·m$^{-1}$·K$^{-1}$] that specifies the value of turbulent thermal conduction of the liquid depending on the Reynolds number; for the conditions in question, $k_\tau$ = 0.016 W·m$^{-1}$·K$^{-1}$.

Formula (27) follows the well-known solution for the temperature, which is set at the boundary of two semi-infinite rods at the moment of their contact [54] (p. 401), with the following assumptions: (1) the molecular thermal conductivity of a liquid is negligibly small compared to its turbulent thermal conductivity, and (2) the turbulent thermal conductivity is a power function of the Reynolds number:

$$\lambda_\tau = k_\tau \mathrm{Re}^{2/3} \tag{32}$$

where $\lambda_\tau$ is the turbulent thermal conductivity [W·m$^{-1}$·K$^{-1}$].

Expression (29) is a linear interpolation of the dependence of the subcooling effect coefficient $\xi$ on water subcooling in a possible variation range from 0.4 to 0.8 (this range for $\xi$ is justified by the authors by analyzing the frequency of the potential contacts of liquid with the solid surface during oscillations of the liquid–gas interface phases near the EFB temperature).

In the presence of oxide scale on the steel surface, the thermophysical properties of the cooled subsurface layer used in the Formula (27) are calculated as follows (values of all properties are understood at temperature $t_w$):

— density [kg·m$^{-3}$]:

$$\rho_w = \rho_{met}(1 - \psi_{sc}) + \rho'_{sc}\psi_{sc} \tag{33}$$

where $\rho_{met}$ is the density of steel; $\rho'_{sc}$ is the apparent (i.e., including pores) density of oxide scale; and $\psi_{sc}$ is the volume fraction of oxide scale (including pores) in the subsurface layer;

— thermal conductivity [W·m$^{-1}$·K$^{-1}$]:

$$\lambda_w = \left[\frac{1 - \psi_{sc}}{\lambda_{met}} + \frac{\psi_{sc}}{\lambda_{sc}}\right]^{-1} \tag{34}$$

where $\lambda_{met}$ is the thermal conductivity of steel, and $\lambda_{sc}$ is the thermal conductivity of oxide scale;

— true isobaric specific-mass-heat capacity [J·kg$^{-1}$K$^{-1}$]:

$$c_w = c_{met}(1 - \varepsilon_{sc}) + c_{sc}\varepsilon_{sc} \tag{35}$$

where $c_{met}$ is the true isobaric specific-mass-heat capacity of steel, and $\varepsilon_{sc}$ is the mass fraction of oxide scale in the subsurface layer.

For steel, well-known formulas approximating the dependence of thermophysical properties on temperature are used (for example, [55], see Appendix B for the properties of the test plates). For scale, the formulas described in by the authors in [56–59] are taken.

The *End of Transient Boiling temperature* $t_{\mathrm{ETB}}$ [°C] is calculated by the formula:

$$t_{\mathrm{ETB}} = t_s + \Delta t^0_{\mathrm{ETB}} k_w \tag{36}$$

where $t_s$ is the water saturation temperature [°C], and $\Delta t^0_{\mathrm{ETB}}$ is the overheating of the surface at the ETB point for a subcooled water at free convection, i.e., without taking into account the water speed [°C]:

$$\Delta t^0_{\mathrm{ETB}} = 50 s_{sub} - \Delta t_{sub} \tag{37}$$

$s_{sub}$ is the coefficient of the influence of water subcooling on the first critical heat flux at free convection [60] (p. 205):

$$s_{sub} = 1 + 0.065 \frac{\overline{c_f} \Delta t_{sub}}{r_v} \left( \frac{\rho_{fs}}{\rho_{vs}} \right)^{0.8} \tag{38}$$

$\overline{c_f}$ is the average isobaric specific-mass-heat capacity of water in the range from $t_f$ to $t_s$ [J·kg$^{-1}$K$^{-1}$]; $r_v$ is the latent heat of vaporization of water [J·kg$^{-1}$]; $\rho_{fs}$ and $\rho_{vs}$ are the density of water and vapor at the saturation temperature, respectively [kg·m$^{-3}$]; and $k_w$ is the coefficient of the influence of water flow velocity on ETB temperature:

$$k_w = \left( 1 + 1.5 u_f^{2/3} \right)^{1/4} \tag{39}$$

$u_f$ is the water speed relative to the sheet surface [m·s$^{-1}$].

Formula (37) is based on the assumption that the heat-transfer coefficient at the first critical point (i.e., at the ETB point) when the liquid is supercooled remains the same as for a saturated liquid. The validity of this assumption is confirmed, for example, by the experimental data cited in [61,62]. Formula (39) is based on the assumption that the volume vapor content of the near-wall layer in the first critical state (at the ETB point) does not depend on the fluid flow speed.

The *heat flux* at the reference points is calculated on the basis of known methods:

— At the End of Film Boiling (i.e., at the EFB-point), according to Wang-Shi [63]:

$$\mathrm{Nu_{EFB}} = \sqrt{\frac{k(m+1)}{\pi}} \mathrm{Re}_x^{(m+1)/2} \mathrm{Pr}_f \tag{40}$$

where $\mathrm{Nu_{EFB}} = q_{\mathrm{EFB}} x / (\lambda_f \Delta t_{sub})$ and $\mathrm{Re}_x = u_f x / \nu_f$ are the local Nusselt number and the local Reynolds number at the distance $x$ [m] from the beginning of the zone, respectively; $q_{\mathrm{EFB}}$ is the local heat flux at the EFB point [W·m$^{-2}$]; $\lambda_f$, $\nu_f$ and $\mathrm{Pr}_f$ are the thermal conductivity [W·m$^{-1}$·K$^{-1}$], kinematic viscosity [m$^2$·s$^{-1}$] and Prandtl number of the liquid far from the surface, respectively; $u_f$ is the liquid flow speed [m·s$^{-1}$]; $\Delta t_{sub}$ is the subcooling of the liquid [°C] (see (30)); and $k = 0.0055$ and $m = 0.68$ are the parameters of the Wang-Shi model, obtained by them from experimental data.

— At the End of Transition Boiling (at the ETB point), in the Kutateladze-Leont'ev method for Critical Heat Flux [64] (p. 311):

$$q_{\mathrm{ETB}} = 2c_{f0}\varphi_*(1-\varphi_*)r_v u_f s_{sub} \sqrt{\rho_{fs}\rho_{vs}} \tag{41}$$

where $c_{f0}$ is the coefficient of friction between the liquid and the surface (evaluated by [65] (p. 289)); $\varphi_*$ is the vapor content in the near-wall two-phase layer in the first critical state (we evaluated $\varphi_*(1-\varphi_*) \approx 0.17$ based on [64] (p. 312)); $r_v$ is the latent heat of the vaporization of water [J·kg$^{-1}$]; $s_{sub}$ is the water subcooling coefficient (see (38)); and $\rho_{fs}$ and $\rho_{vs}$ are the density of water and vapor at the saturation temperature, respectively [kg·m$^{-3}$].

— at the End of Nucleate Boiling (at the ENB point), according to Isachenko-Kushnyrev [66] (p. 178).

### 2.1.3. Microstructure

Microstructure is used in block 10 of the diagram in Figure 2.

The evolution of the microstructure in the sheet during cooling is predicted from isothermal Time–Temperature-Transformation (TTT) diagrams [67–69] (p. 356), using a step-by-step calculation scheme based on the additivity rule [70–72].

In order to use the TTT diagrams originally presented in graphical form, the authors modified the method of approximation by reference points proposed in [73,74] (p. 43). The essence of such modification is that in addition to the two reference points—at the beginning and at the "nose" of the C-shaped curve—another reference point below the "nose" is introduced (Figure 9). This makes it possible to significantly improve the approximation

accuracy, especially in the lower part of the C-curve, i.e., at temperatures predominantly related to the bainite transformation.

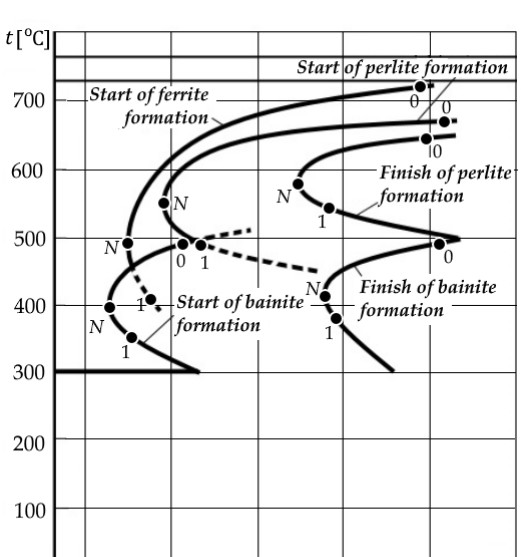

**Figure 9.** Example of a TTT-diagram showing a 3-point approximation scheme. Diagram shows the reference points taken in the approximation for each C-curve: 0—upper, N—"nose", 1—lower.

The method of 3-point approximation proposed by the authors allows each of the five basic C-shaped curves of the austenite isothermal transformation (start of ferrite formation, start and finish of pearlite formation, start and finish of bainite transformation) to be described by a single-parameter function of the following form:

$$y = w^c e^{c(1-w)} \tag{42}$$

where $y$ is the function of the logarithm of isothermal holding time $\tau$ (as an independent coordinate of the TTT diagram), $w$ is the function of temperature $t$ (as a dependent coordinate) and $c$ is the parameter to be calculated by the following formula:

$$c = \frac{\ln y_1}{1 - w_1 + \ln w_1} \tag{43}$$

The functions of the coordinates in Formulas (42) and (43) are as follows:

$$y = \frac{S - S_0}{S_N - S_0} \tag{44}$$

$$w = \frac{U - U_0}{U_N - U_0} \tag{45}$$

$$y_1 = \frac{S_1 - S_0}{S_N - S_0} \tag{46}$$

$$w_1 = \frac{U_1 - U_0}{U_N - U_0} \tag{47}$$

where $S$ means the logarithm of time:

$$S_0 = \ln \tau_0, \ S_N = \ln \tau_N, \ S_1 = \ln \tau_1, \ S = \ln \tau \tag{48}$$

and $U$ means the inverse temperature:

$$U_0 = 1000/t_0, \ U_N = 1000/t_N, \ U_1 = 1000/t_1, \ U = 1000/t \tag{49}$$

Subscript "0" refers to the upper reference point (at the beginning of the C-curve), "*N*" refers to the reference point at the C-curve "nose" and 1 refers to the lower reference point (see Figure 9).

Within the above scheme of using TTT diagrams to predict the steel microstructure during cooling, the kinetics of isothermal austenite decomposition into ferrite and pearlite is calculated using the Kolmogorov–Johnson–Mehl–Avrami (KJMA) equation [75] (p. 128, 496), the bainite transformation via the Austin–Rickett equation [76,77] and the martensite transformation via the Koistinen–Marburger equation [78].

### 2.1.4. Temperature Distribution across the Thickness of the Sheet

Temperature distribution across the thickness of the sheet is used in block 11 of the Diagram in Figure 2.

A procedure for numerically solving the one-dimensional unsteady thermal conductivity equation for a flat, metal plate with oxide scale on both wide surfaces, with boundary conditions of the third kind, has been implemented.

The design scheme is shown in Figure 10. The designations in this figure are as follows: $H$ is the sheet thickness, including oxide scale; $h_{sb}$ and $h_{st}$ are the thickness of the oxide scale layer on the bottom and top surface, respectively; $h_m$ is the thickness of the metal body without oxide scale; $n_{sb} \geq 3$ is the quantity of nodes of the computation grid inside the bottom oxide scale layer (the quantity of nodes inside the top oxide scale layer should also be not less than 3); $n_m$ is the quantity of nodes of the computation grid inside the metal body; $n$ is the total quantity of nodes across the thickness of the sheet with oxide scale (the value of $n$ is determined automatically, based on the thickness of the elementary layer $\Delta x$ specified in the initial data); $i$ is the number of the current node; $\delta_b$ and $\delta_t$ are the thickness of internal elementary layer of the bottom and top oxide scale, respectively; and $\Delta x$ is the thickness of the internal elementary layer of the metal body.

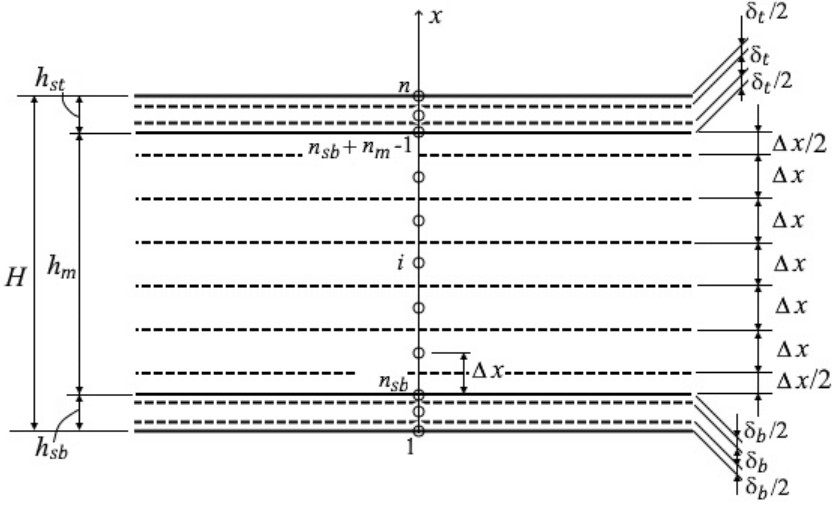

**Figure 10.** The design scheme for numerical calculation of the sheet temperature ($x$ is the axis along the thickness of the sheet). Designations are in the text.

The finite-difference equations to be solved at any time step, except the initial one (i.e., at time $\tau > 0$), are as follows:

— Inside the metal body ($n_{sb} + 1 \leq i \leq n_{sb} + n_m - 2$):

$$c_i^k \rho_i^k \frac{t_i^{k+1} - t_i^k}{\Delta \tau} = \lambda_i^k \cdot \frac{t_{i+1}^{k+1} - 2t_i^{k+1} + t_{i-1}^{k+1}}{(\Delta x)^2} + q_{Vi} \tag{50}$$

where $t_i^k$ is the temperature in the $i$-th node at the $k$-th point in time [°C]; $\Delta \tau$ is the time step between the $k$-th and $k + 1$-th points in time [s]; $c_i^k$, $\rho_i^k$ and $\lambda_i^k$ are the values of the properties

of steel in the *i*-th node at the *k*-th point in time: isobaric specific-mass-heat capacity [J·kg$^{-1}$·K$^{-1}$], density [kg·m$^{-3}$] and thermal conductivity [W·m$^{-1}$·K$^{-1}$], respectively; and $q_{Vi}$ is the volumetric latent heat capacity of phase transformations in steel [W·m$^{-3}$] (values for austenite transformation are taken from [79]);

— Inside the bottom scale layer ($2 \leq i \leq n_{sb} - 1$):

$$c_{sci}^k \rho_{sci}^k \frac{t_i^{k+1} - t_i^k}{\Delta \tau} = \lambda_{sci}^k \cdot \frac{t_{i+1}^{k+1} - 2t_i^{k+1} + t_{i-1}^{k+1}}{\delta_b^2}, \tag{51}$$

where $c_{sci}^k$, $\rho_{sci}^k$ and $\lambda_{sci}^k$ are the values of the properties of oxide scale in the *i*-th node at the *k*-th point in time: isobaric specific-mass-heat capacity [J·kg$^{-1}$·K$^{-1}$], density [kg·m$^{-3}$] and thermal conductivity [W·m$^{-1}$·K$^{-1}$], respectively;

— Inside the top scale layer ($n_{sb} + n_m \leq i \leq n - 1$):

$$c_{sci}^k \rho_{sci}^k \frac{t_i^{k+1} - t_i^k}{\Delta \tau} = \lambda_{sci}^k \cdot \frac{t_{i+1}^{k+1} - 2t_i^{k+1} + t_{i-1}^{k+1}}{\delta_t^2} \tag{52}$$

— Conjugation conditions between the bottom scale layer and the metal ($i = n_{sb}$):

$$\frac{\lambda_i^k}{\Delta x} \left( t_{i+1}^{k+1} - t_i^{k+1} \right) = \frac{\lambda_{sci}^k}{\delta_b} \left( t_i^{k+1} - t_{i-1}^{k+1} \right) \tag{53}$$

— Conjugation conditions between the top scale layer and the metal ($i = n - n_m + 1$):

$$\frac{\lambda_i^k}{\Delta x} \left( t_i^{k+1} - t_{i-1}^{k+1} \right) = \frac{\lambda_{sci}^k}{\delta_t} \left( t_{i+1}^{k+1} - t_i^{k+1} \right) \tag{54}$$

— Boundary conditions at the bottom scale surface ($i = 1$):

$$\alpha_b^k \left( t_i^{k+1} - t_{ab}^{k+1} \right) - \frac{\lambda_{sci}^k}{\delta_b} \left( t_{i+1}^{k+1} - t_i^{k+1} \right) = -c_{sci}^k \rho_{sci}^k \frac{\delta_b}{2} \left( \frac{t_i^{k+1} - t_i^k}{\Delta \tau} \right) \tag{55}$$

where $\alpha_b^k$ is the heat-transfer coefficient at the bottom surface of the sheet at the *k*-th point in time [W·m$^{-2}$·K$^{-1}$], and $t_{ab}^{k+1}$ is the ambient temperature at the bottom surface at the *k* + 1-th point in time [°C];

— Boundary conditions at the top scale surface ($i = n$):

$$\alpha_t^k \left( t_n^{k+1} - t_{at}^{k+1} \right) + \frac{\lambda_{sci}^k}{\delta_t} \left( t_i^{k+1} - t_{i-1}^{k+1} \right) = -c_{sci}^k \rho_{sci}^k \frac{\delta_t}{2} \left( \frac{t_i^{k+1} - t_i^k}{\Delta \tau} \right). \tag{56}$$

where $\alpha_t^k$ is the heat-transfer coefficient at the top surface at the *k*-th point in time [W·m$^{-2}$·K$^{-1}$], and $t_{at}^{k+1}$ is the ambient temperature at the top surface at the *k*+1-th point in time [°C].

The system of *n* Equations (50)–(56) is solved using an implicit finite-difference scheme via the Thomas (tridiagonal matrix) algorithm [80] (p. 83). The step of the calculation grid over the thickness of the sheet $\Delta x$ is about 0.1 mm, and the basic time step $\Delta \tau$ is about 0.1 s (additionally divided by the borders between the designed cooling zones on the top and bottom surfaces of the sheet).

## 2.2. Experimental Studies

### 2.2.1. Experimental Procedure

In order to check the adequacy and to adapt the model, the authors used the data from previously conducted temperature measurements across the thickness of the steel plate during processing in a roller-quenching machine (RQM) of NKMZ design [81].

The technique of the experimental studies is described in detail in [82,83]. It consisted of the application of a measuring complex in the form of a test plate with embedded thermocouples, connected to the data-collection and recording system (Figure 11).

The experiments involved two test plates of structural carbon steel 45 (standard chemical composition in weight is C = 0.42 . . . 0.5%, Si = 0.17 . . . 0.37%, Mn = 0.5 . . . 0.8%) with dimensions of 30 × 2000 × 4000 mm. Four thermocouples were embedded in each of them (see scheme in Figure 12): in the first plate, there were three thermocouples in the middle of the width (one at the top, bottom and center of the thickness) and one at the top at a distance of 200 mm from the lateral edge; in the second plate, there were two thermocouples in the middle of the width (one at the top and bottom) and two at a distance of 200 mm from the lateral edge (one at the top and bottom). The depths of the surface thermocouples were 3 and 2.5 mm from the top and bottom surfaces, respectively. A total of two series of five tests each were carried out. The first plate was used in test Nos. 1–5, and the second plate in test Nos. 6–10.

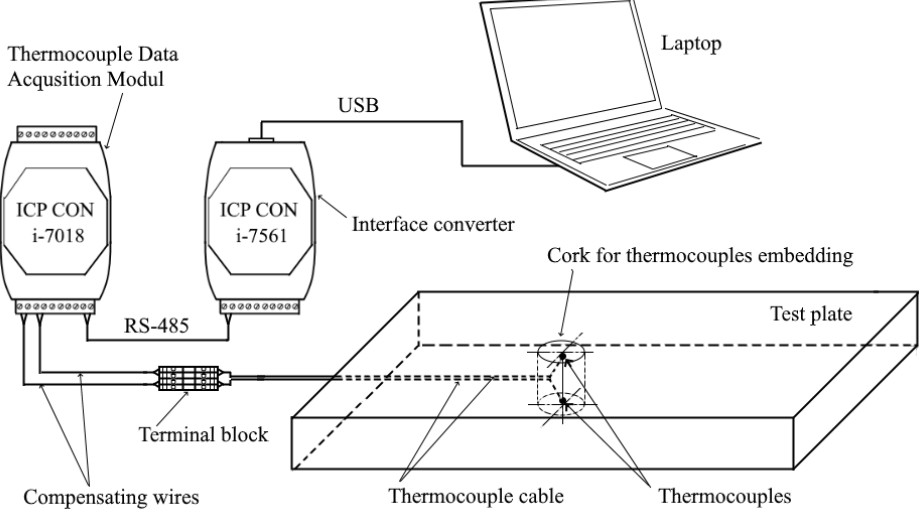

**Figure 11.** Schematic of the measuring complex.

The scheme of each test involved charging a cold plate with thermocouples into a heating furnace from the RQM side (i.e., through the furnace outlet window), heating it to a preset temperature, discharging it from the furnace at a constant speed and cooling it in the roller-quenching machine. The heating temperature and RQM operating regime in each test are given in Table 1.

The signals from the thermocouples were recorded continuously during the whole process of heating the test plate in the furnace, its transportation and subsequent cooling in the RQM. As an example, Figure 13 shows the plots of temperature measured at different control points of the plate during a single test (No. 2). The temperature measurements in the furnace (i.e., for the time from $t_0$ to $t_1$) were used to calculate the thickness of the oxide scale layer on the plate surface. The measurement data after the plate was discharged from the furnace (after $t_1$) was compared with the results of the cooling model calculation.

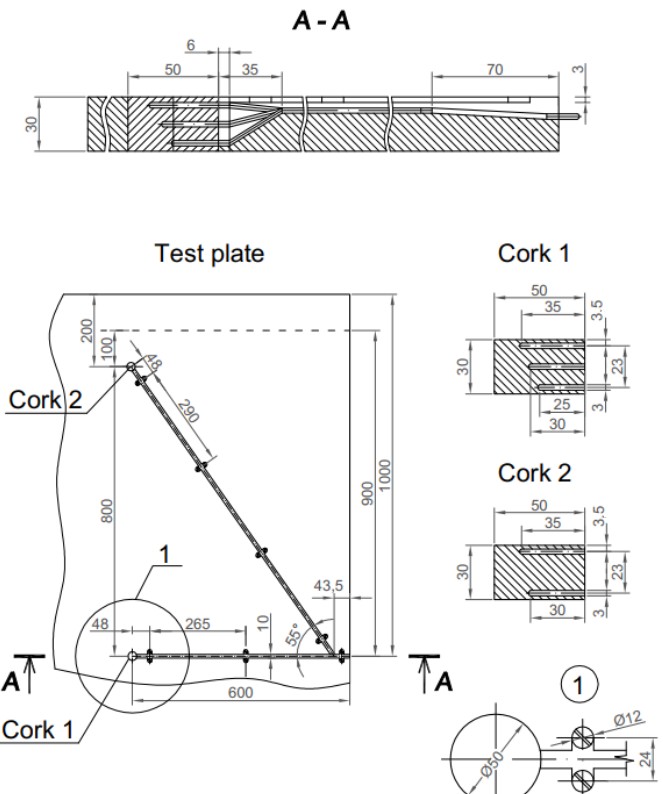

**Figure 12.** Thermocouple embedding scheme (dimensions are in mm). Of the five channels prepared, four were used in each test plate (details in the text).

**Table 1.** Setpoint heating temperature and RQM operating regime in tests.

| Test No. | Heating Temperature [°C] | Plate Speed [ms$^{-1}$] | Water Temperature [°C] | Water Flow Rates by RQM Zone *: Top (above the Line) and Bottom (below the Line) [m$^3$h$^{-1}$] | | | | | | |
|---|---|---|---|---|---|---|---|---|---|---|
| | | | | H1 | H2-3 | L1-2 | L3 | L4 | L5 | L6 |
| 1 | 850 | 0.12 | 23 | 530 | 290 | 110 | 110 | 110 | 0 | 0 |
| | | | | 510 | 510 | 190 | 190 | 190 | 0 | 0 |
| 2 | 950 | 0.11 | 23 | 310 | 290 | 110 | 110 | 110 | 0 | 0 |
| | | | | 510 | 510 | 190 | 190 | 190 | 0 | 0 |
| 3 | 950 | 0.11 | 23 | 0 | 290 | 110 | 110 | 110 | 75 | 0 |
| | | | | 0 | 490 | 190 | 190 | 190 | 115 | 0 |
| 4 | 950 | 0.11 | 23 | 310 | 100 | 110 | 110 | 110 | 70 | 0 |
| | | | | 510 | 180 | 190 | 190 | 190 | 115 | 0 |
| 5 | 950 | 0.11 | 23 | 310 | 0 | 110 | 110 | 110 | 0 | 0 |
| | | | | 510 | 0 | 190 | 190 | 190 | 0 | 0 |
| 6 | 950 | 0.11 | 34 | 0 | 120 | 130 | 130 | 90 | 130 | 130 |
| | | | | 0 | 210 | 210 | 210 | 160 | 210 | 210 |
| 7 | 950 | 0.11 | 34 | 310 | 0 | 130 | 110 | 110 | 75 | 0 |
| | | | | 310 | 200 | 190 | 153 | 170 | 115 | 0 |
| 8 | 950 | 0.09 | 34 | 310 | 0 | 130 | 0 | 110 | 75 | 0 |
| | | | | 310 | 200 | 190 | 0 | 170 | 115 | 0 |

**Table 1.** *Cont.*

| Test No. | Heating Temperature [°C] | Plate Speed [ms⁻¹] | Water Temperature [°C] | Water Flow Rates by RQM Zone *: Top (above the Line) and Bottom (below the Line) [m³h⁻¹] | | | | | | |
|---|---|---|---|---|---|---|---|---|---|---|
| | | | | H1 | H2-3 | L1-2 | L3 | L4 | L5 | L6 |
| 9 | 950 | 0.09 | 33 | 310 | 0 | 130 | 0 | 110 | 75 | 0 |
| | | | | 210 | 200 | 190 | 0 | 170 | 115 | 0 |
| 10 | 980 | 0.11 | 32 | 310 | 100 | 0 | 0 | 90 | 110 | 75 |
| | | | | 410 | 175 | 0 | 0 | 155 | 170 | 115 |

* The designations of RQM zones are as follows: type (H—High-intensive, L—Low-intensive) and ordinal number (H2-3 and L1-2 are paired zones with twin banks).

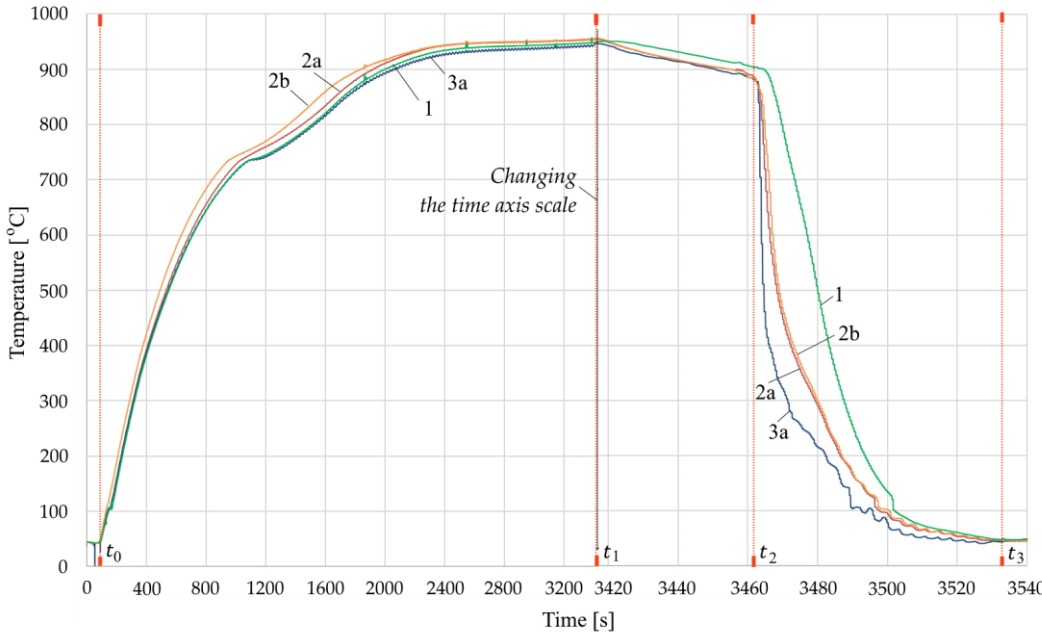

**Figure 13.** Plots of temperature changes, measured using thermocouples in test No. 2 during heating and cooling. Designations of the curves: 1—at the center of the thickness in the middle of the plate width; 2a—at the top in the middle of the width; 2b—at the top at the edge; and 3a—at the bottom in the middle of the width. Designations of the key moments of time: $t_0$—entry into the furnace; $t_1$—exit from the furnace; $t_2$—entry into the first zone of RQM; $t_3$—exit from RQM.

2.2.2. Estimation of Oxide Scale Thickness on the Surface of the Test Plates

We accepted the diffusion mechanism of oxide scale growth, according to which the rate of its mass increase is inversely proportional to the mass of already-formed scale [84] (p. 49):

$$\frac{dY}{d\tau} = \frac{K^2}{2Y} \tag{57}$$

where $Y$ is the scale mass per unit surface [kg·m⁻²]; $\tau$ is time [s]; and $K$ is the oxidation rate constant [kg·m⁻²s⁻⁰·⁵].

The integration of (57) gives a parabolic equation of scale growth [85]:

$$Y^2 = Y_0^2 + K^2\tau \tag{58}$$

where $Y_0$ is the scale mass per unit surface [kg·m⁻²] at $\tau = 0$ s.

Differential Equation (57) is non-linear, because the oxidation rate constant changes in time with temperature change [86] (p. 59):

$$K = A \cdot \exp\left(-\frac{B}{T}\right) \tag{59}$$

where $T$ is an absolute temperature in degrees Kelvin, and $A$ and $B$ are material parameters. For steel 45, the values of these parameters were taken according to [87] (p. 146) as $A = 11.41$ kg·m$^{-2}$s$^{-0.5}$ and $B = 8274$ K.

To numerically solve differential Equation (57), we used Euler's method via an explicit finite-difference scheme:

$$Y_{i+1} = Y_i + \frac{A^2}{2Y_i} \cdot \exp\left(-\frac{2B}{T_i}\right)\Delta\tau \tag{60}$$

where $Y_i$ and $Y_{i+1}$ are the specific mass of oxide scale at the beginning and at the end of the $i$-th time interval [kg·m$^{-2}$]; $T_i$ is the surface temperature measured at the beginning of the $i$-th time interval, K; and $\Delta\tau$ is a measurement cycle, which is equal to 0.07 s.

The transition from specific mass to scale thickness was carried out as follows:

$$h_i = \frac{Y_i}{(1 - \eta_{sc})\, \rho_{sc}} \tag{61}$$

where $h_i$ is the oxide scale thickness corresponding to its specific mass $Y_i$ [m]; $\eta_{sc}$ is the porosity of oxide scale in fractions of one; and $\rho_{sc}$ is its true density [kg·m$^{-3}$]. For furnace scale, formed when heated to 850–980 °C, we assumed $\eta_{sc} = 0.15$ [88] and $\rho_{sc} = 5500$ kg·m$^{-3}$ [56].

Using the described procedure, we calculated the thickness of oxide scale at the exit of the furnace before each test. In this case, given that visually, no coarse scale was observed on the top surface of the plates after treatment in RQM, the initial thickness of the oxide layer upon loading into the furnace was taken to be zero (more precisely, 1 micron in order to avoid dividing by zero in the Formula (60) for the first cycle of measurements). The calculated values of the scale thickness at the exit of the furnace are shown in Table 2.

**Table 2.** Measured temperature and calculated scale thickness at the exit of the furnace on the top (above the line) and the bottom (below the line) surface of the plate *.

| Test No. | Total Heating Time [min-sec] | Plate Temperature [°C] | Oxide Scale Thickness [μm] |
|---|---|---|---|
| 1 | 47′10″ | 853 | 47 |
| | | 845 | 42 |
| 2 | 55′27″ | 956 | 113 |
| | | 948 | 103 |
| 3 | 46′16″ | 957 | 105 |
| | | 950 | 97 |
| 4 | 43′57″ | 956 | 105 |
| | | 949 | 95 |
| 5 | 43′30″ | 956 | 103 |
| | | 950 | 94 |
| 6 | 56′28″ | 962 | 104 |
| | | 958 | 98 |
| 7 | 52′23″ | 958 | 123 |
| | | 954 | 116 |
| 8 | 51′48″ | 957 | 123 |
| | | 951 | 117 |

**Table 2.** *Cont.*

| Test No. | Total Heating Time [min-sec] | Plate Temperature [°C] | Oxide Scale Thickness [μm] |
|---|---|---|---|
| 9 | 45′29″ | 958 | 110 |
| | | 950 | 104 |
| 10 | 60′43″ | 982 | 147 |
| | | 976 | 138 |

* In the tests, in which there were two thermocouples on one surface, the final values of temperature and scale thickness for the entire surface are taken by averaging the data for both points.

## 3. Results

According to the authors' cooling model, calculations were performed for the conditions of each of the ten tests. The following values of the basic input data (in addition to the data presented in Tables 1 and 2 and Figure 12) were taken:

— The design and layout characteristics of RQM, according to the technical documentation of the equipment manufacturer (basic characteristics are given in [31,81,89]);
— The thermophysical properties of the test plate as a function of temperature, according to the formulas [55] for medium-carbon steel (see Appendix B);
— The thermophysical properties of oxide scale as a function of temperature, according to the authors' formulas [56–59];
— The oxide scale thickness was assumed to be constant throughout the cooling period of each test and equal to the values given in Table 2.

The results of the calculations in comparison with the corresponding experimental data are presented in the form of graphs of temperature changes over time. For example, Figures 14–16 compare the experimental and calculated graphs for test No. 3, No. 5 and No. 10, respectively. Appendix A shows similar graphs for the remaining seven tests.

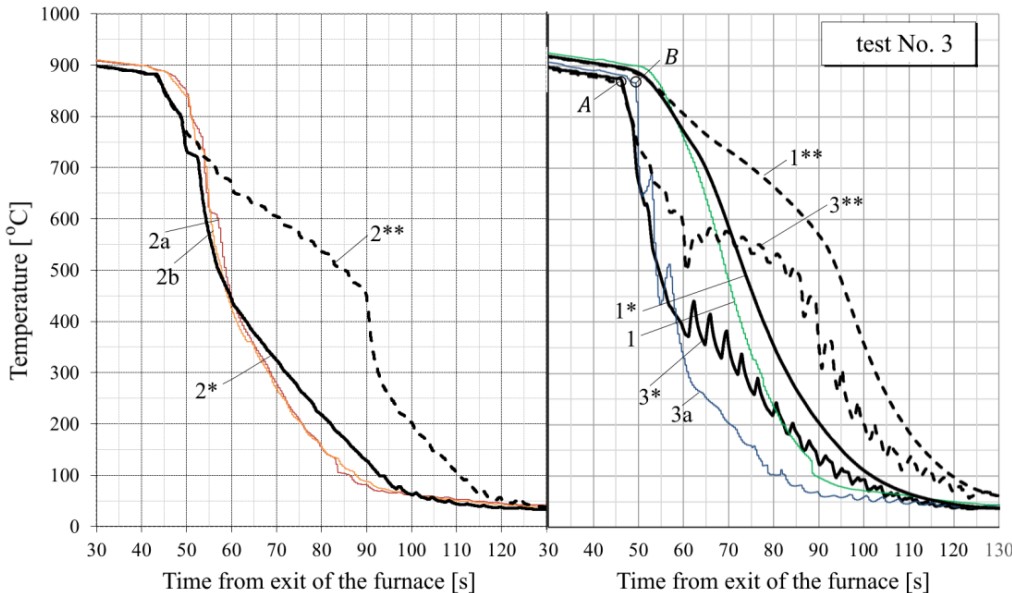

**Figure 14.** Graphs of cooling in test No. 3. Numbers without an asterisk denote experimental curves according to the thermocouple data: 1—at the center of the thickness in the middle of the plate width; 2a—at the top in the middle of the width; 2b—at the top at the edge; 3a—at the bottom in the middle of the width. Numbers with one asterisk indicate calculated curves for the corresponding control points across the thickness of the plate: 1* and 1**—at the center of the sheet thickness with regard to oxide scale and without regard to oxide scale, respectively; 2* and 2**—at a depth of 3 mm from the top surface with regard to oxide scale and without regard to oxide scale, respectively; 3* and 3**—at

a depth of 2.5 mm from the bottom surface with regard to oxide scale and without regard to oxide scale, respectively. Point A is the first inflection of the calculated temperature graph, corresponding to the beginning of water cooling from below, and point B is the same in the test (its lag for several seconds is probably due to the thermal inertia of the thermocouple).

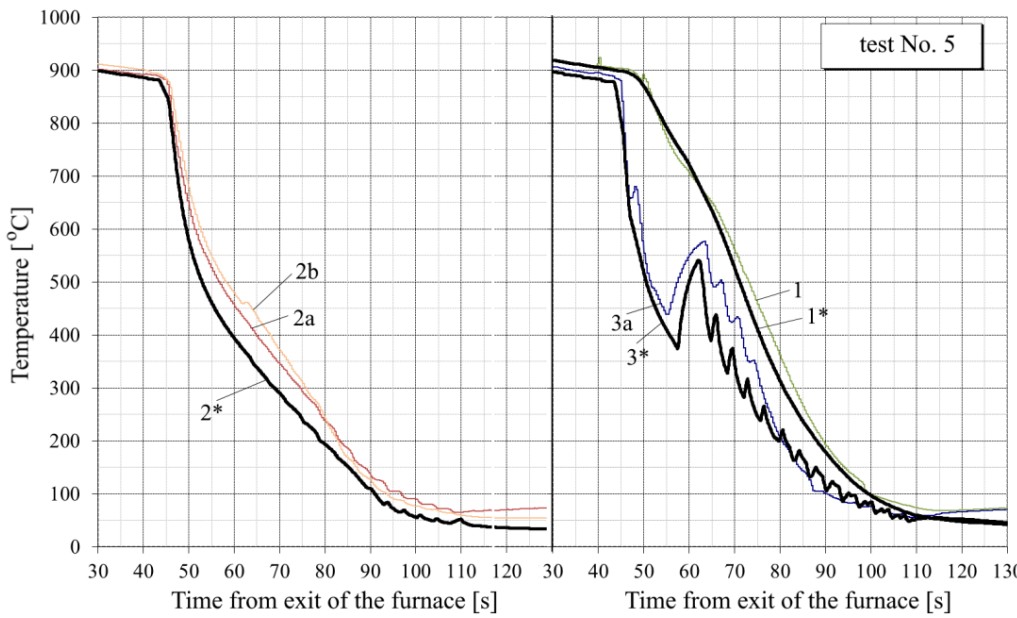

**Figure 15.** Graphs of cooling in test No. 5. The notations are the same as in Figure 14.

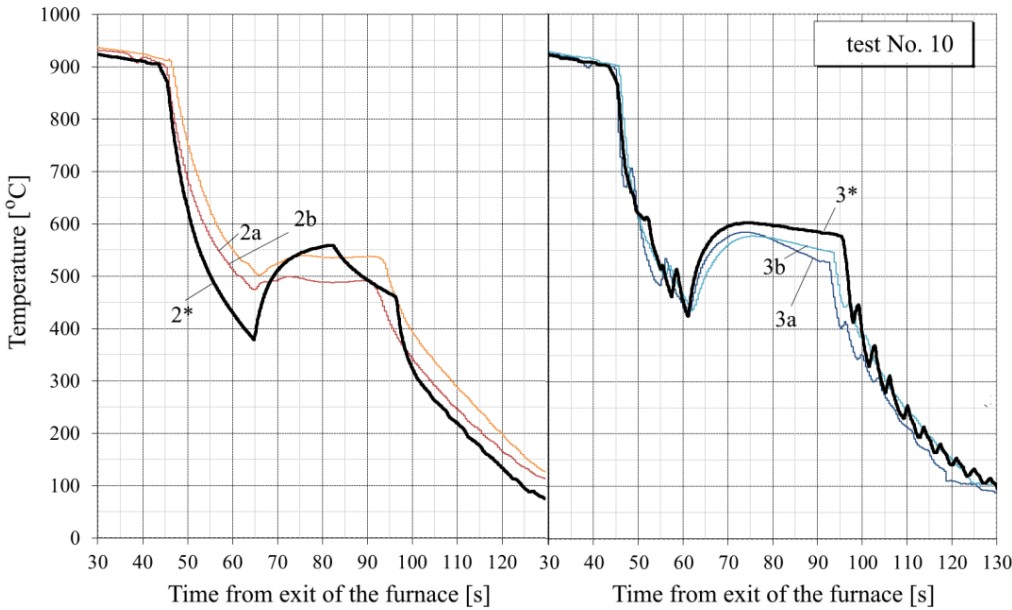

**Figure 16.** Graphs of cooling in test No. 10. Numbers without an asterisk denote experimental curves according to the thermocouple data: 2a—at the top in the middle of the width; 2b—at the top at the edge; 3a—at the bottom in the middle of the width; 3b—at the bottom at the edge. Numbers with one asterisk indicate calculated curves for the corresponding control points across the thickness of the plate: 2*—at a depth of 3 mm from the top surface; 3*—at a depth of 2.5 mm from the bottom surface.

The analysis of the data obtained shows the following.

(1)     For all tests, the calculated temperature at the RQM outlet corresponds to the measured temperature, with deviations not exceeding 10 °C. Therefore, the total heat loss of the sheet is taken into account correctly, which indirectly confirms the adequacy of

the calculation of the first and second critical surface temperatures corresponding to the changes in the film, transition and nucleate boiling regimes (i.e., the EFB and ETB temperature in Figure 8).

(2) In the tests with interrupted cooling (Nos. 5, 8, 9 and 10—see Figures 15, 16, A6 and A7), the calculated temperature graphs repeat the characteristic changes in the course of the experimental curves. This suggests that the sizes of the characteristic zones of jet cooling, as well as the parameters of water spreading over the top and bottom surfaces of the sheet, are correctly taken into account.

(3) The degree of closeness of the calculated and experimental curves remains approximately at the same level at significantly different water temperatures (from 23 °C in tests Nos. 1–5 to 32–34 °C in tests Nos. 6–10), which indicates the correctness of taking this factor into account in the model.

(4) Oxide scale thickness on the plate surface is the main parameter of the model which defines its agreement with the experiment. For comparison, Figure 14 also shows the graphs for test No. 3, calculated without taking oxide scale into account (dotted curves marked by the numbers with two asterisks). It can be seen that in this case, there are very rough discrepancies between the calculated and experimental data.

(5) In tests Nos. 2, 3, 6 and 7 (Figures 14, A2, A4 and A5), in the area of surface thermocouple readings below 400–450 °C, the calculated temperature is much higher than the experimental one. In our opinion, this is due to the assumption made in the simulation of constant thickness of the scale during the entire cooling period. This assumption is generally not true, because during accelerated cooling, oxide scale can crack and be removed (partially or completely) from the sheet surface by water jets. At the same time, the influence of scale depends on the water boiling regime: at high temperatures corresponding to film and transient boiling, oxide scale, as a rule, increases the intensity of heat transfer to the surface; at stable nucleate boiling, on the contrary, reduces it [90]. Therefore, if oxide scale is removed from the sheet surface after stable nucleate boiling is achieved, it is accompanied by an increase in the intensity of cooling. In spray cooling, stable nucleate boiling of water usually begins at 200–250 °C at the surface, which, at the corresponding heat flux values, approximately corresponds to a temperature of 400–450 °C at a depth of 2.5–3 mm (for a plate 30 mm thick). This explains the above-mentioned overestimation of the calculated temperature in the noted experiments with surface thermocouple readings below 400–450 °C. This effect is also confirmed by model calculations. For example, in Figure 14, it is seen that at indications of surface thermocouples above 400–450 °C, the slope angle of the experimental plots is close to the slope angle of the design graphs with scale, and below this boundary, to the slope angle of design graphs without scale.

(6) In practically all cases, there is a "lag" for 1–3 s of the experimental curves from the calculated ones at the very beginning of intensive cooling (see, for example, points A and B in Figure 14). This, most likely, can be explained by the thermal inertia of thermocouples [91], which manifests, to the greatest extent, as a sharp change in metal temperature.

(7) To quantify the "proximity" of the experimental and calculated graphs, the value of the average cooling rate at a certain temperature interval was used. Table 3 summarizes the average cooling rate in the three typical temperature ranges: 800–400 °C, 400–200 °C and 200–100 °C. The value in each cell of this table is obtained by averaging over all experiments. It can be seen that calculated cooling rates are, in general, somewhat lower than experimentally determined (on average, by 12–20% at different temperature intervals). Higher cooling rates in the experiments (than in the simulation) can be explained by the factors mentioned above: the thermal inertia of thermocouples (see item 6) and scale removal from the plate surface during cooling in RQM (see item 5). In this case, if the thermal inertia of thermocouples affects only the "apparent" cooling rate, the reduction in the scale layer affects the actual intensity of heat transfer.

**Table 3.** Average cooling rate in characteristic temperature intervals.

| Nature of the Data | Reference Coordinate by Sheet Thickness * | Average Cooling Rate [°C/s] in Temperature Range | | |
|---|---|---|---|---|
| | | 800–400 °C | 400–200 °C | 200–100 °C |
| Experiment (thermocouples) | top | 27.5 | 14.4 | 8.0 |
| | center | 23.7 | 20.3 | 11.4 |
| | bottom | 42.2 | 14.3 | 7.9 |
| Calculation (model) | top | 29.4 | 11.0 | 8.4 |
| | center | 21.2 | 16.3 | 9.0 |
| | bottom | 32.2 | 11.9 | 6.5 |
| Deviation (calculation minus experiment) | top | 1.8 | −3.5 | 0.4 |
| | center | −2.5 | −4.0 | −2.4 |
| | bottom | −10.0 | −2.4 | −1.5 |
| | averaged over three coordinates ** | −3.6 (−12%) | −3.3 (−20%) | −1.1 (−13%) |

* Top—3 mm from the top surface, bottom—2.5 mm from the bottom surface. ** Percentages are relative to experimental values.

## 4. Discussion

The obtained results confirm that the main factor that introduces uncertainty into the process of the accelerated cooling of metal in production conditions is oxide scale on its surface. And the point is not that scale, especially peeled scale, distorts the readings of the workshop pyrometers. Much more important is the fact that oxide scale changes the real intensity of cooling because it shifts the boiling curve. When cooling a surface with oxide scale, the boiling curve, in general, is a composition of two curves (Figure 17) [90]: at the initial stage, it follows the boiling curve for the surface covered with a continuous layer of scale, and then, due to scale cracking, it shifts in the direction of the boiling curve for a clean surface. Therefore, in the presence of scale, in general, two more reference points are added to the boiling curve:

SHX—Start sHift due to oXide layer and
EHX—End sHift due to oXide layer.

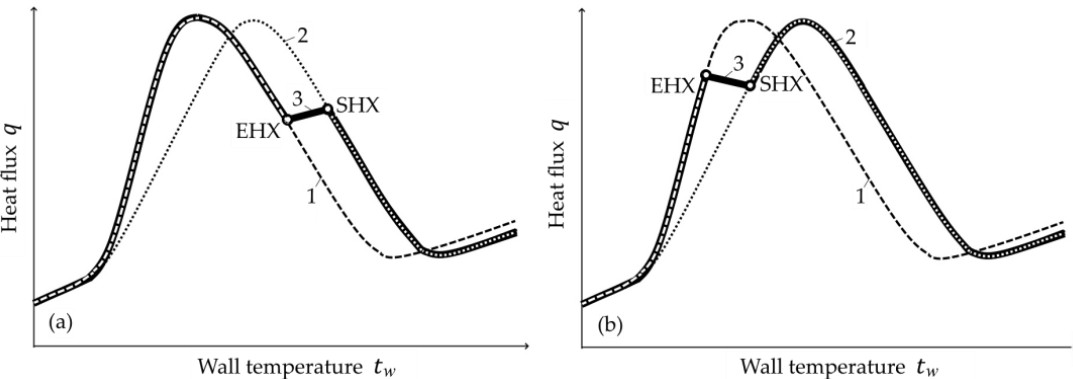

**Figure 17.** Schematic of boiling curve formation during water-jet cooling on the surface covered by an oxide scale. Boiling curve symbols: 1 (dashed)—on the clear metal surface without oxide scale; 2 (dotted)—on the surface with a hard scale; 3 (solid)—in cases of scale cracking. SHX and EHX—the reference points associated with oxide scale (transcript in the text). (**a**) an example of when scale cracking occurs at the transient boiling regime (in this case, oxide scale increases heat flux); (**b**) an example of when scale cracking occurs at the nucleate boiling regime (in this case, oxide scale decreases heat flux).

The moment at which this "jump" from one boiling curve to the other occurs (see Figure 17) will determine both the course of the cooling process and the final temperature of the metal. Unfortunately, the cracking and removal of oxide scale from the sheet surface is very difficult to predict because it is a random event, which creates inevitable uncertainty when modeling cooling over a wide temperature range. Moreover, in practice, the initial thickness of the surface scale layer is also usually unknown.

This predetermines two main directions of further research: (1) predicting the thickness of scale during TMCP processing, taking into account its initial thickness, the strength of adhesion with the base metal and the dynamics of oxide thickness changes in the cooling unit; and (2) using the thickness of the oxide scale layer as the main parameter for the on-line adaptation of the temperature model in the ACS of accelerated cooling and quenching units.

## 5. Conclusions

Comparison with experimental data shows that the new version of the authors' mathematical model of jet cooling adequately takes into account the main design and technological factors determining the sheet temperature; this includes water consumption by the sections, the sequence of their inclusion, the temperature of cooling water and the thickness of scale on the surface. This makes it possible to recommend a new version of the model for use in control systems for accelerated sheet-metal cooling in a wide temperature range, up to full hardening below 100 °C. It is shown that the main disturbing factor that introduces uncertainty into the simulation results under real conditions is the thickness of the oxide scale layer on the sheet surface and the random nature of its removal in the process of jet cooling. Therefore, the rated thickness of the scale layer should be considered the main parameter for adapting the process of controlling the sheet temperature.

**Author Contributions:** Conceptualization, E.B. and Y.B.; methodology, E.B. and Y.B.; software E.B.; validation, E.B.; formal analysis, E.B.; investigation, E.B.; resources, E.B. and Y.B.; data curation, E.B.; writing—original draft preparation, E.B.; writing—review and editing, E.B. and Y.B.; visualization, E.B.; supervision, Y.B.; project administration, Y.B.; funding acquisition, E.B. All authors have read and agreed to the published version of the manuscript.

**Funding:** This research received no external funding.

**Data Availability Statement:** The data presented in this study are available on request from the corresponding author.

**Acknowledgments:** The authors are grateful to D.O. Kozlenko for developing the software user service of the cooling model, and to A.L. Ostapenko for helpful advice when discussing the results of this work.

**Conflicts of Interest:** The authors declare no conflict of interest.

**Appendix A**

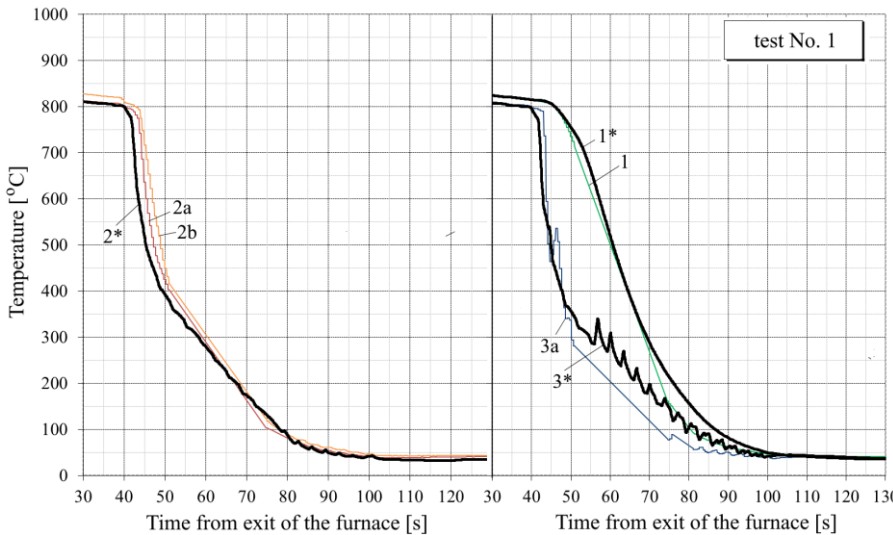

**Figure A1.** Graphs of cooling in test No. 1. Numbers without an asterisk denote experimental curves according to the thermocouple data: 1—at the center of the thickness in the middle of the plate width; 2a—at the top in the middle of the width; 2b—at the top at the edge; 3a—at the bottom in the middle of the width. Numbers with one asterisk indicate calculated curves for the corresponding control points across the thickness of the plate: 1*—at the center of the sheet thickness; 2*—at a depth of 3 mm from the top surface; 3*—at a depth of 2.5 mm from the bottom surface.

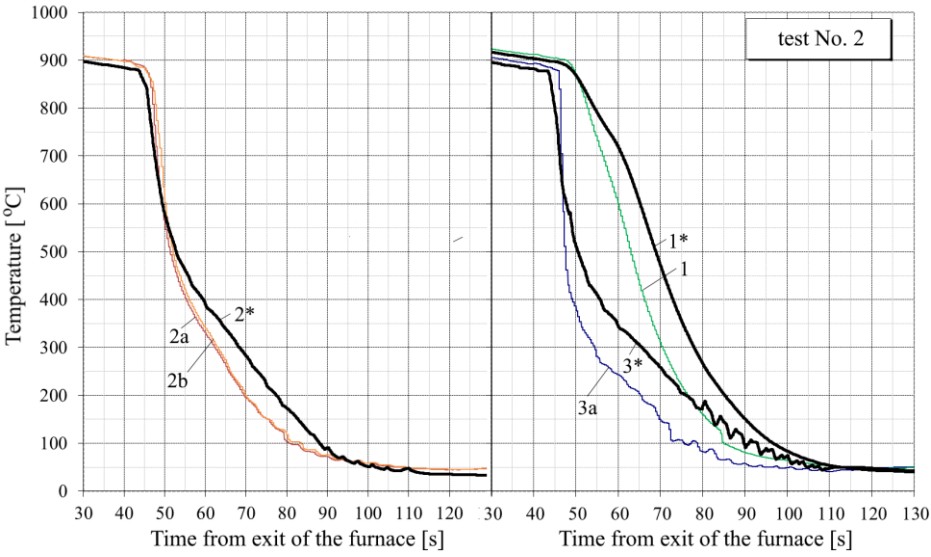

**Figure A2.** Graphs of cooling in test No. 2. The notations are the same as in Figure A1.

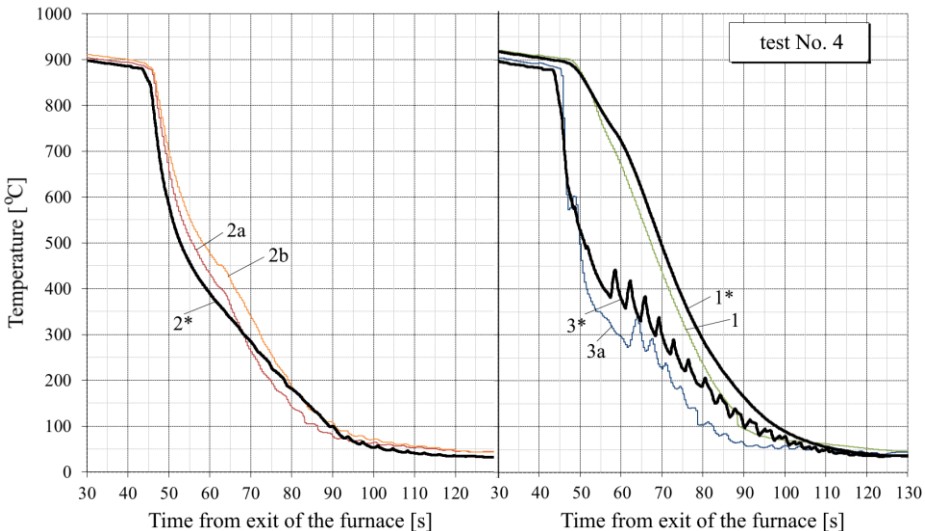

**Figure A3.** Graphs of cooling in test No. 4. The notations are the same as in Figure A1.

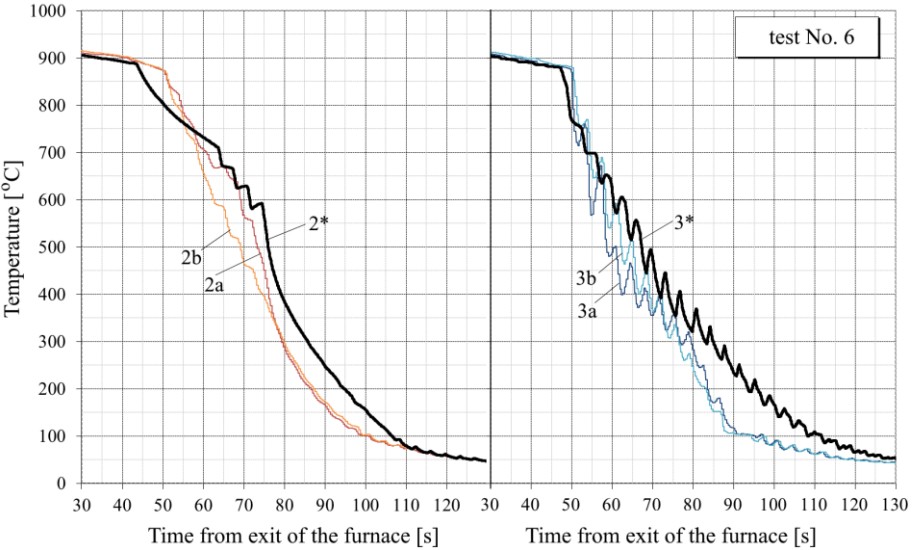

**Figure A4.** Graphs of cooling in test No. 6. Numbers without an asterisk denote experimental curves according to the thermocouple data: 2a—at the top in the middle of the width; 2b—at the top at the edge, 3a—at the bottom in the middle of the width; 3b—at the bottom at the edge. Numbers with one asterisk indicate calculated curves for the corresponding control points across the thickness of the plate: 2*—at a depth of 3 mm from the top surface; 3*—at a depth of 2.5 mm from the bottom surface.

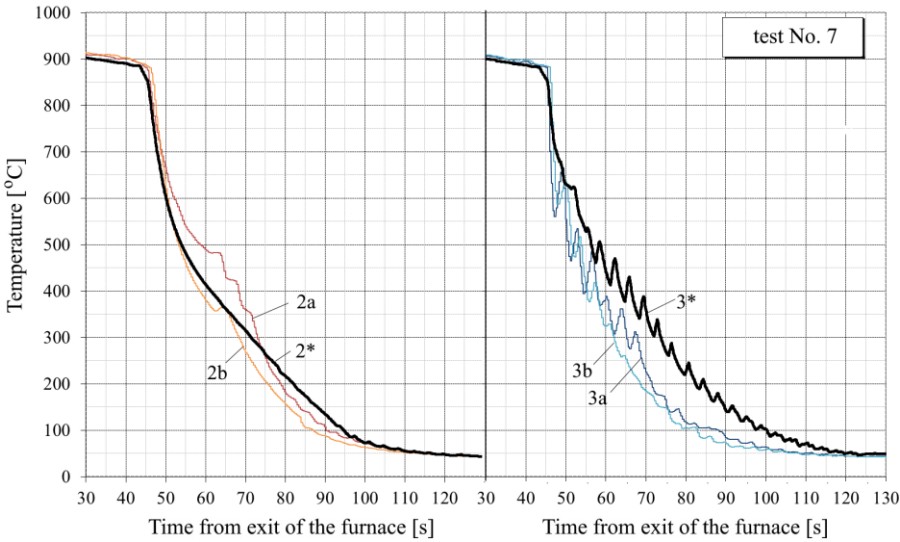

**Figure A5.** Graphs of cooling in test No. 7. The notations are the same as in Figure A4.

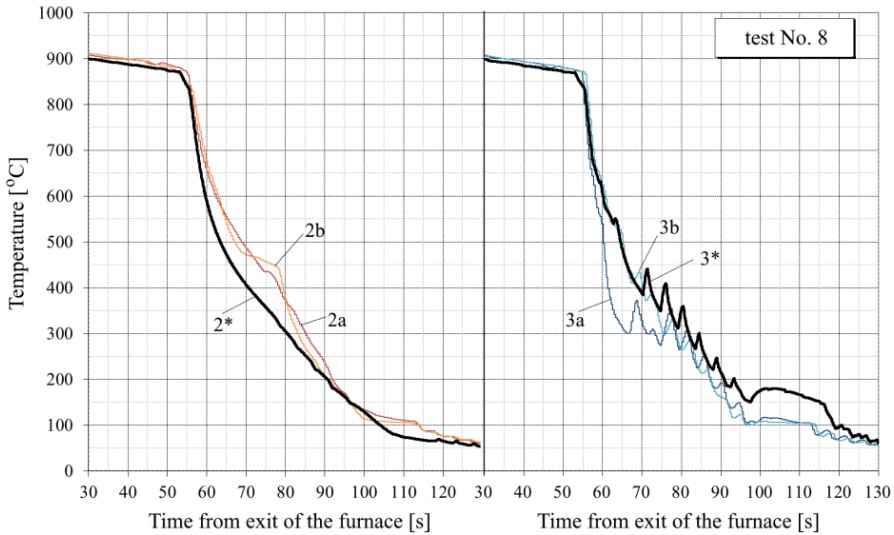

**Figure A6.** Graphs of cooling in test No. 8. The notations are the same as in Figure A4.

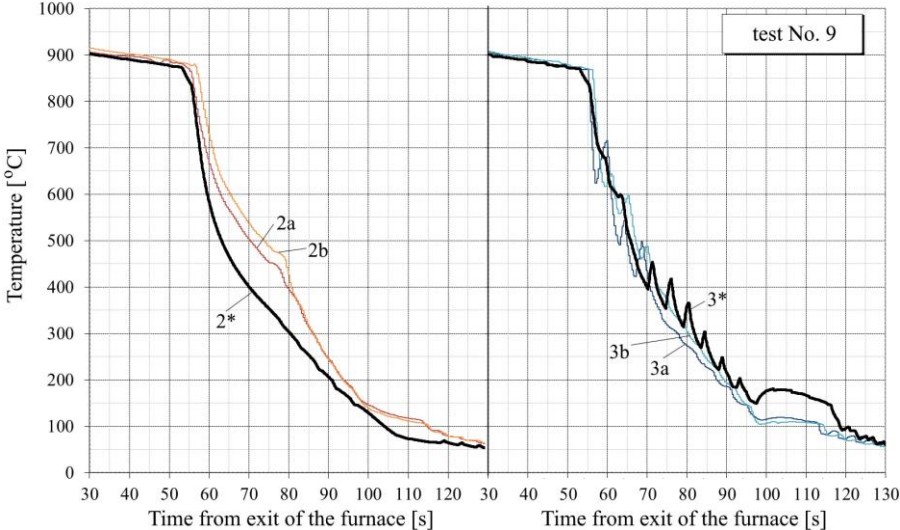

**Figure A7.** Graphs of cooling in test No. 9. The notations are the same as in Figure A4.

**Appendix B**

The temperature dependence of the thermal physical properties for the material of the test plates (steel 45) according to the approximation formulae from [55] for medium-carbon steel:

- Thermal conductivity [W·m$^{-1}$·K$^{-1}$]:

$$\lambda = 55.94 - \frac{31.28}{\cosh(2.85\cdot 10^{-3}(t - 935))} \tag{A1}$$

- True isobaric specific-mass-heat capacity [J·kg$^{-1}$·K$^{-1}$]:

$$c = 481.5 + 0.2t + 812.2e^{-a|t-768|} \tag{A2}$$

where $a$ = 0.0099 for $t \leq 768$ °C and $a$ = 0.0261 for $t > 768$ °C.

- Density [kg·m$^{-3}$]:

$$\rho = \frac{7850}{1 + 3\alpha(t - 20)} \tag{A3}$$

where $\alpha$ is the mean coefficient of linear thermal expansion in a temperature range from 20°C to $t$ [K$^{-1}$]:

$$\alpha = 10^{-6}\left[10.7 + 6\cdot 10^{-3}t - \frac{2.9}{\cosh\left(7.6\cdot 10^{-5}(t - 905)^2\right)}\right] \tag{A4}$$

In the above formulae, $t$ is the steel temperature in °C, and cosh is the hyperbolic cosine.

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
