# Peer review of "Validation of the Cooling Model for TMCP Processing of Steel Sheets with Oxide Scale Using Industrial Experiment Data"

_jmmp, doi:10.3390/jmmp6040078_

Round 1

Reviewer 1 Report

The research is of interest for practical applications in metallurgical production and when using heat treatment of steel. The authors have demonstrated an original approach for modeling the thermomechanical process in order to predict and control the characteristics of rolled steel.

As a remark, it should be noted:

1.  A general note: when describing approaches in calculations with references to the published works used, it is still desirable to provide in the text the main formulas and dependencies referred to by the authors.

2. The main content of the model is based on the behavior description of the cooling medium (water), including its geometry, aggregation state, mobility, and thermo-physical properties. All this is given in the conditions of a cooled fixed steel plate. In Fig. 1, rollers are provided, which means the movement of rolled sheet. Did the authors consider a variant of their model in the case of water-cooling of a moving plate at a certain speed? Do the authors admit such conditions with a mobile steel plate and what basic factors will play their role to a greater extent (a more elongated area of the water jet impact spot, cooling intensity, etc.)?

3. When analyzing the microstructure, the authors relied on the well-known structural characteristics of quenching diagrams (according to the references given). As a recommendation, I would like to see a visual confirmation of the structural-phase analysis at those processing points that the authors describe (metallography, X-ray). It is no secret, which even small structural fluctuations can affect the rate of physical and chemical processes associated with oxidation.

4. The authors claim that the main uncontrollable and unpredictable factor is the oxide film on the steel plate. This film is considered as part of the initial structural state of the steel plate before quenching. Nevertheless, additional oxidation of the steel plate during interaction with water is not excluded that the authors did not take into account. That is, the scale amount will change not only in plate volume (width and thickness), but also in the time of the cooling process, which also affects the temperature parameters.

Author Response

Dear Reviewer,

We appreciate your comments, which prompted us to revise and allowed to improve the manuscript. Below we provide a response to each of your comments (under the corresponding numbers). All relevant changes in the text of the manuscript are highlighted in blue. 

  1. We have added the main formulas of other authors used in the model: for the thermophysical properties of steel (Appendix B), for heat flux ate the EFB and ETB points (see formulas (40 and (41)).
  2. Indeed, in the current version of the model, most of the parameters are calculated under the assumption that the sheet is stationary. The movement of the sheet is taken into account in the sense that each of its cross-sections at different times passes through different cooling zones as it passes through the cooling unit. We agree with the reviewer that the movement of the sheet affects the cooling conditions, especially the shape of the spot. And we have already begun work on taking this factor into account, but at the moment we do not mention it among the two main directions for the development of the model (see the end of the “Discussion” section).

  3. Unfortunately, we were not able to perform a microstructural analysis of the test sheets, since they had to be left for subsequent experiments.
  4. We agree with your comment. Taking this comment into account, we have adjusted the text of directions for further research (line 669).

Please take note that we have also corrected the English slightly, and in the revised version of the manuscript the grammatical and sentence corrections are highlighted in yellow.

Kind regards,

Emmanuil Beygelzimer, author

Reviewer 2 Report

The manuscript can be accepted by addressing the following comments:

1) The motivation of this work is not clear.

2) Please add some recent references in the introduction.

3) There is no conclusion section in the manuscript. 

4) Check for various grammatical and sentence errors to improve the quality of the manuscript. 

Author Response

Dear Reviewer,

We appreciate your comments, which prompted us to revise and allowed to improve the manuscript. Below we provide a response to each of your comments (under the corresponding numbers). All relevant changes in the text of the manuscript are highlighted in blue.

  1. We have adjusted the end of the Introduction section and clarified the goal of the work.
  2. We have added four recent references in the introduction (No. 2, 8, 12 and 18 in the revised reduction).
  3. Conclusion section added.
  4. In the revised version of manuscript, grammatical and sentence corrections are highlighted in yellow.

Kind regards,

Emmanuil Beygelzimer, author